# Actin stress fiber organization promotes cell stiffening and proliferation of pre-invasive breast cancer cells

Sandra Tavares[1], André Filipe Vieira[2,3,*], Anna Verena Taubenberger[4,*], Margarida Araújo[1,*], Nuno Pimpao Martins[1], Catarina Brás-Pereira[1], António Polónia[2,3,5], Maik Herbig[4], Clara Barreto[1], Oliver Otto[4], Joana Cardoso[1,6], José B. Pereira-Leal[1,6], Jochen Guck[4], Joana Paredes[2,3,7] & Florence Janody[1]

Studies of the role of actin in tumour progression have highlighted its key contribution in cell softening associated with cell invasion. Here, using a human breast cell line with conditional Src induction, we demonstrate that cells undergo a stiffening state prior to acquiring malignant features. This state is characterized by the transient accumulation of stress fibres and upregulation of Ena/VASP-like (EVL). EVL, in turn, organizes stress fibres leading to transient cell stiffening, ERK-dependent cell proliferation, as well as enhancement of Src activation and progression towards a fully transformed state. Accordingly, EVL accumulates predominantly in premalignant breast lesions and is required for Src-induced epithelial overgrowth in *Drosophila*. While cell softening allows for cancer cell invasion, our work reveals that stress fibre-mediated cell stiffening could drive tumour growth during premalignant stages. A careful consideration of the mechanical properties of tumour cells could therefore offer new avenues of exploration when designing cancer-targeting therapies.

[1] Instituto Gulbenkian de Ciência, Rua da Quinta Grande 6, P-2780-156 Oeiras, Portugal. [2] Epithelial Interactions in Cancer group, Instituto de Investigação e Inovação em Saúde (i3S), Universidade do Porto, Rua Alfredo Allen, 208, 4200–135 Porto, Portugal. [3] Cancer Genetics group, Instituto de Patologia e Imunologia Molecular da Universidade do Porto (Ipatimup), Rua Júlio Amaral de Carvalho 45, 4200–135 Porto, Portugal. [4] Biotechnology Center, Technische Universität Dresden, Tatzberg 47/49, 01307 Dresden, Germany. [5] Department of Pathology, Ipatimup Diagnostics, Ipatimup, Rua Dr Roberto Frias s/n, 4200–465 Porto, Portugal. [6] Ophiomics—Precision Medicine, Rua Cupertino de Miranda 9, lote 8, 1600–513 Lisboa, Portugal. [7] Department of Pathology, Faculty of Medicine, University of Porto, Alameda Prof. Hernaâni Monteiro, 4200–319 Porto, Portugal. * These authors contributed equally to this work. Correspondence and requests for materials should be addressed to F.J. (email: fjanody@igc.gulbenkian.pt).

Breast cancer is a major cause of death in women worldwide[1]. The multistep process of breast cancer progression results from the acquisition of genetic and epigenetic alterations in oncogenes and tumour suppressor genes, which confer growth and/or survival advantage to mammary cells. Subsequent molecular alterations may convert these premalignant cells into malignant ones, with invasive and metastatic abilities[2].

The viral non-receptor tyrosine kinase v-Src and its cellular homologue c-Src are the most investigated proto-oncogenes, implicated in many aspects of tumour development, including proliferation, survival, adhesion, migration, invasion and metastasis[3]. Src protein levels and, to a greater extent, Src protein kinase activity are frequently elevated in malignant and non-malignant breast tissues and significantly associated with decreased survival of breast cancer patients[4,5].

Src induces tumour metastasis mainly by reducing adhesiveness and by regulating the actin cytoskeleton[3]. The semi-flexible polymers of filamentous actin (F-actin), which are assembled from monomeric actin subunits (G-actin), exert or resist forces to drive a large number of cellular processes, including changes in cell shape, cell mobility, cytokinesis and intracellular transport. In addition, actin filaments translate external forces into biochemical signalling events that guide cellular responses[6]. This has been largely studied in the context of tumour invasion and malignancy, where mechanical signals from the tumour microenvironment impact the metastatic cascade[7]. In turn, metastatic breast cells have lower stiffness than their healthy counterparts, which is largely determined by the cytoskeleton[8,9]. To perform these different functions, actin filaments organize into distinct architectures through the control of a multitude of actin-binding proteins (ABPs) strongly conserved between species[10]. Ena/VASP (enabled/vasodilator stimulated phosphoprotein) family proteins, including protein-enabled homologue (Mena), vasodilator-stimulated phosphoprotein (VASP) and Ena-VASP-like (EVL), associate with barbed ends of actin filaments. They appear to have different effects on F-actin, including favouring actin filament elongation through their anti-capping activity, inhibiting the formation of branched actin networks and promoting F-actin bundling[11]. Accordingly, Ena/VASP family members have distinct effects on cancer cell migration and metastasis. While EVL suppresses cell migration[12], the Mena variant Mena (INV) drives invasion, intravasation and metastasis[13]. Other ABPs inhibit actin polymerization or stabilize actin filaments. In addition, some ABPs organize actin filaments into higher-order networks, by cross-linking actin filaments, or use F-actin as a scaffold, physical support or track, to promote contractility and generate tension[10].

We have previously reported that the pro-growth function of Src is controlled by the actin cytoskeleton in *Drosophila* epithelia[14]. In this report, we investigate the role of F-actin in supporting the expansion of cancer precursors downstream of Src. Using a breast cell line with conditional Src induction, we demonstrate that prior to cells acquiring malignant features, they undergo a transient stress-fibre-dependent stiffening state leading to cell proliferation and the progression towards a fully transformed state.

## Results

**Src sustains proliferation prior to inducing migration**. The main transformation events that occur upon overactivation of the Src oncogene can be studied in cell culture, using the mammary MCF10A epithelial cell model with conditional Src induction, which contains a fusion between v-Src and the ligand-binding domain of the Oestrogen Receptor (ER-Src), inducible with tamoxifen (TAM) treatment[15,16]. ER-Src cells treated with the

vehicle EtOH, but not TAM-treated MCF10A cells carrying an empty vector (PBabe; Supplementary Fig. 1), showed basal levels of phosphorylated ER-Src (ER-pSrc), indicating that ER-Src displays some degree of leakiness. However, treating ER-Src cells with TAM potentiated ER-pSrc levels (Fig. 1a), which increased in a stepwise manner during the 36 h of TAM treatment (Fig. 1b). In addition, TAM treatment triggered the phosphorylation of endogenously expressed Src (pSrc) (Fig. 1a). In two-dimensional (2D) cultures, this untransformed cell line undergoes morphological transformation features 36 h after TAM treatment with progressive cell detachment (Fig. 1c and Supplementary Movie 2). This is in contrast to ER-Src cells treated with the vehicle EtOH (Fig. 1c and Supplementary Movie 1) or PBabe cells treated with TAM (Supplementary Fig. 1). In three-dimensional (3D) cultures of reconstituted basement membrane, TAM but not EtOH treatment triggered transformed features, characterized by the extrusion of cells from the spherical acinar-like structure that invaded the Matrigel 45 h after treatment (compare Supplementary Movies 3 and 4).

In the presence of serum and growth factors, TAM-treated ER-Src cells did not display an increase in the percentage of cells in S-phase of the cell cycle over EtOH-treated cells during the 36 h of treatment (Supplementary Fig. 2). Consistent with a requirement for epidermal growth factor (EGF) to support proliferation of untransformed MCF10A cells[17], in the absence of EGF, the EtOH-treated ER-Src cell population showed a progressive decrease in their proliferation rate over 36 h of treatment. Cells that had been treated with TAM, however, had a significant proliferation advantage over EtOH-treated cells, starting 12 h after TAM treatment (Fig. 1d). Because cell tracking showed no significant difference in velocity between EtOH- and TAM-treated ER-Src cells during the first 12 h or between 36 and 48 h of treatment (Supplementary Fig. 2), we conclude that TAM-treated ER-Src cells acquire self-sufficiency in growth properties before migrating abilities.

**Src sustains cell proliferation in part via ERK activation**. The acquisition of self-sufficiency in growth properties is unlikely to result from increased activity of the mediators of Hippo signalling, Yes-associated protein (YAP) and transcriptional co-activator with PDZ-binding motif (TAZ), as we could not observe YAP/TAZ being enriched in the nuclei of ER-Src cells grown in the absence of serum and growth factors during the first 24 h of TAM treatment (Supplementary Fig. 3). Moreover, microarray analysis indicated that the YAP/TAZ target genes *connective tissue growth factor (CTGF), ankyrin repeat domain 1 (ANKRD1), cysteine-rich angiogenic inducer 61 (CYR61), baculoviral IAP repeat containing 5 (BIRC5), AXL receptor tyrosine kinase (AXL), inhibin alpha subunit (INHA)* and *collagen type VIII alpha 1 chain (COL8A1)* are not upregulated in TAM-treated ER-Src cells[18]. Finally, quantitative PCR with reverse transcription 12 and 24 h after TAM treatment did not reveal higher CTGF or ANKRD1 expression (Supplementary Fig. 3). As extracellular signal-regulated kinases (ERK) have been shown to contribute to EGF-independent growth of MCF10A cells[19] and are important contributors of tumour growth[20], we tested the role of ERK in Src-sustained proliferation. Accordingly, in the absence of serum and growth factors, the levels of the phosphorylated form of ERK (pERK) were significantly higher in ER-Src cells 12 h after TAM treatment (Fig. 1e). Moreover, these cells upregulated Cyclin D1 (Fig. 1f), a known regulator of G1 to S phase progression[20]. ERK is required to potentiate Cyclin D1 expression in these cells, as co-treatment with PD184352, an inhibitor of the Mitogen-activated protein kinase kinase (MEKi), strongly reduced pERK levels in both EtOH- and TAM-treated cells at 12 h (Fig. 1g), but

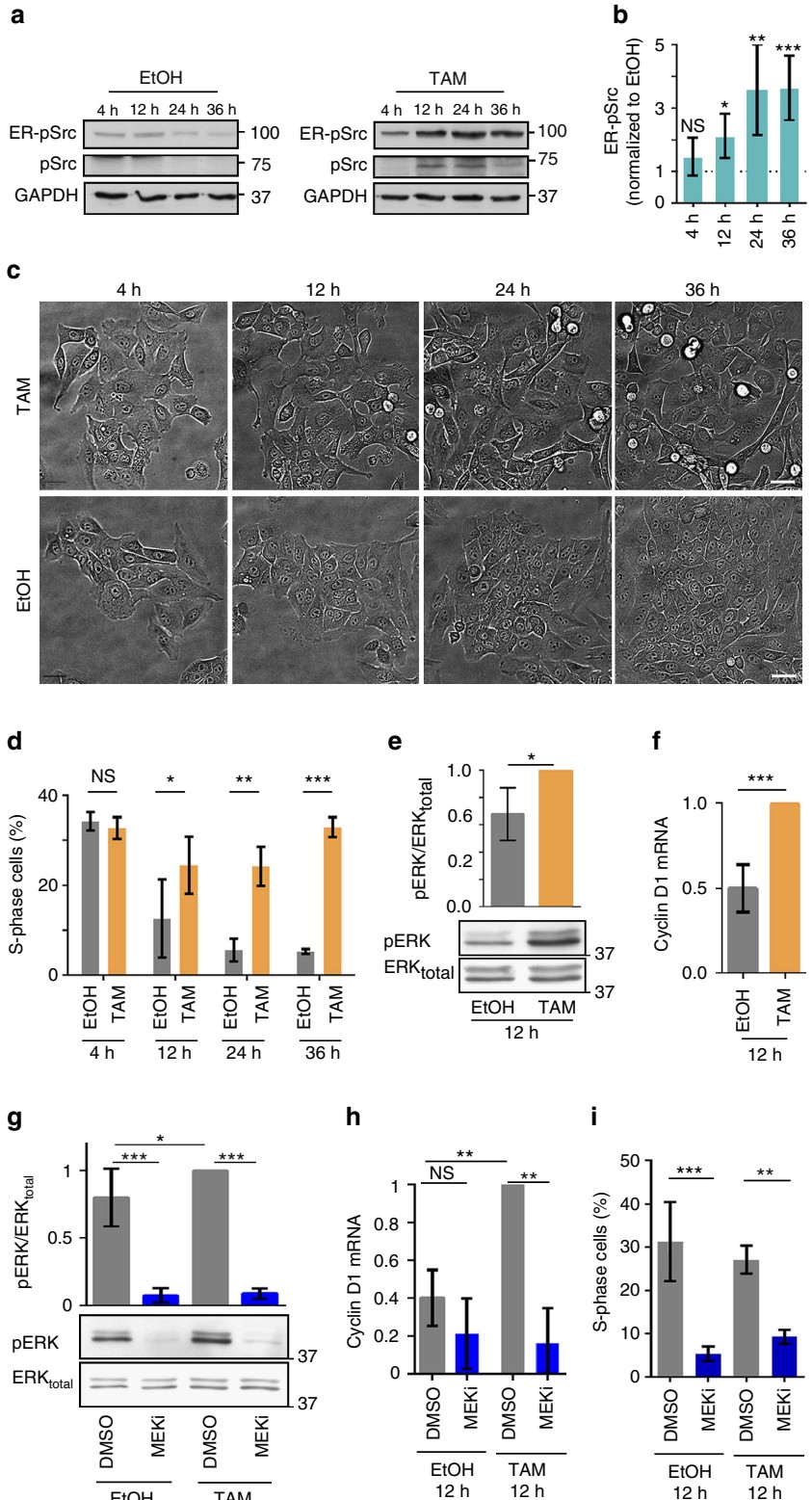

prevented the upregulation of Cyclin D1 exclusively in TAM-treated cells (Fig. 1h). Furthermore, in the presence of serum and growth factors, co-treatment of ER-Src cells with MEKi and EtOH or TAM for 12 h reduced the percentage of S-phase, compared to cells treated with DMSO (Fig. 1i). We conclude that Src activation sustains cell proliferation by potentiating ERK activation and Cyclin D1 upregulation.

**Src transiently boosts F-actin assembly and cell stiffening**. We then investigated the effect of Src activation on F-actin when cells acquire proliferative abilities. Strikingly, TAM-treated ER-Src cells transiently accumulated basal actin fibres, mainly observed 12 h after Src induction (Fig. 2a and Supplementary Fig. 1). Scoring of the percentage of cells with high basal F-actin signals showed that only TAM treatment for 12 h significantly increased

the fraction of cells with high basal F-actin levels (Fig. 2b). Western blot analysis indicated that Src induction promoted a significant increase in the ratio between F- and G-actin 12 h after TAM treatment (Fig. 2c,d). Since total actin levels remained constant during the transformation process (Fig. 2e), these transient changes in F-actin are apparently due to an increase in actin polymerization.

Higher absolute levels of F-actin or stress fibre formation have been associated with increased cell stiffness, whereas a reduction in F-actin or stress fibre disassembly correlates with lower cell stiffness[8,21,22]. Accordingly, atomic force microscopy (AFM)-based indentation measurements revealed that ER-Src (but not PBabe) cells were significantly stiffer between 4 and 14 h after TAM treatment compared to EtOH-treated cells (Fig. 2f and Supplementary Fig. 1). In contrast, between 36 and 38 h, when TAM-treated ER-Src cells displayed morphological transformation features, they were more deformable than control cells (Fig. 2f). Similarly, ER-Src cells isolated from TAM-treated 3D cultures showed a transient increase in cell stiffness at 12 h (Fig. 2g). However, cell stiffness was similar between TAM- and EtOH-treated cells at 36 h (Fig. 2g). This may be due to the fact that phenotypic transformation could take longer in 3D cultures (Supplementary Movie 4). In accordance with AFM experiments, real-time deformability cytometry (RT-DC) measurements on suspended cells confirmed that ER-Src cells were stiffer 12 h after TAM treatment (Fig. 2h). Moreover, ER-Src cells treated with TAM for 12 h accumulated higher levels of phospho-myosin light chain (pMLC) compared to cells treated with EtOH for the same period of time (Fig. 2i). Taken together, we conclude that Src-induced cellular transformation involves a transient increase in actin fibres associated with cell stiffening and the acquisition of self-sufficiency in growth properties.

**Common ABPs are deregulated by Src and in DCIS.** To further characterize the mechanisms leading to the transient increase in F-actin and cell stiffness, we searched for ABPs involved in building these F-actin structures. Microarray profiling identified 35 ABPs deregulated during the 36 h of Src-induced cellular transformation (Fig. 3a)[18]. To test which of these ABPs would be relevant for breast cancer progression, we analysed the expression signature related to the actin cytoskeleton in premalignant atypical ductal hyperplasia (ADH), ductal carcinoma *in situ* (DCIS) and invasive ductal carcinoma (IDC) (Fig. 3b). Using cancer expression data sets available from the GEO database, we collected microarray data for 255 normal breast tissues and 903 neoplastic breast lesions, from which the available information relative to the diagnosis, histological grade and ER status was extracted (Supplementary Table 1). After normalization of the raw data and statistical comparison of breast lesions with normal breast tissues, we sorted all genes significantly deregulated

(Supplementary Data 1). As expected, classification of genes into functional categories using Pathway Express indicated that most of the significantly affected biological processes were associated with cancer (Fig. 3b). Strikingly, 'regulation of the actin cytoskeleton' was among the top 10 pathways significantly deregulated in both premalignant lesions (ADH and DCIS). In contrast, while IDCs upregulated many ABPs previously implicated in cancer cell mobility[23] (Supplementary Data 2), the category 'regulation of the actin cytoskeleton' had a lower impact factor in these malignant lesions relative to other misregulated pathways or was not being significantly affected (Fig. 3b). In addition, classification of these ABPs into functional categories based on their role in F-actin dynamics (Supplementary Data 2) showed that only malignant lesions were enriched for inhibitors of polymerization (Supplementary Fig. 4). Thus, major alterations in the expression of ABPs are predominantly found in premalignant ADH and DCIS.

Among the 27 ABPs deregulated in premalignant ADH and/or DCIS, *EVL*, Actin-related protein 3 (*ACTR3*), Actin-related protein 2/3 complex subunit 5-like protein (*ARPC5L*), Dystronin (*DST*), FH1/FH2 domain-containing protein 3 (*FHOD3*) and Tropomyosin beta chain (*TPM2*) were also deregulated in TAM-induced ER-Src cells (Fig. 3a)[18]. These ABPs may therefore affect the expansion of TAM-treated ER-Src cells by regulating the transient accumulation of actin fibres.

**ABPs deregulated by Src impact tissue growth in *Drosophila*.** To investigate the effect of these ABPs in Src-induced proliferation, we first screened for those that affect the pro-growth function of Src in *Drosophila*, as this organism contains only one family member for each of these ABPs (Supplementary Table 2), reducing the risk of gene redundancy. As expected, distal wing discs overexpressing *Drosophila* Src oncogene at 64B (Src64B) together with the Caspase inhibitor p35 and Green fluorescent protein (GFP) using the the Nubbin-Gal4 (Nub-Gal4) driver were significantly bigger than control discs expressing GFP only (Fig. 4a,b). Strikingly, replacing UAS-GFP by a UAS construct expressing double-strand RNA (dsRNA) directed against EVL/Ena in Nub > Src/p35 wing discs fully suppressed tissue overgrowth (Fig. 4a,b and Supplementary Table 2). In contrast, knocking down ACTR3/Arp3, ARPC5L/Arpc5, DST/Shot, FHOD3/Fhos or TPM2/Tm2 enhanced the overgrowth of Nub > Src/p35-expressing wing discs (Supplementary Table 2 and Supplementary Fig. 5). In the converse experiments, overexpressing EVL/Ena strongly enhanced Src-induced tissue overgrowth (Fig. 4a,b and Supplementary Table 2), while overexpressing ACTR3/Arp3 or DST/Shot or an activated form of Fhos deleted of the conserved C terminal basic cluster (EGFP-Fhos-βB) reduced the overgrowth of these tissues (Supplementary Table 2 and Supplementary Fig. 5). These observations suggest that EVL/Ena promotes Src/p35-induced tissue overgrowth, while

**Figure 1 | ERK sustains proliferation of TAM-treated ER-Src cells at 12 h.** (**a**) Western blots on protein extracts from ER-Src cells treated with EtOH or TAM for 4, 12, 24 or 36 h, blotted with anti-pSrc, which reveals ER-pSrc or endogenous pSrc or anti-GAPDH. (**b**) Ratio of ER-pSrc levels between TAM- and EtOH-treated ER-Src cells for the same time points, normalized to GAPDH. (**c**) Images by phase contrast microscopy of ER-Src cells, treated with TAM or EtOH for 4, 12, 24 or 36 h. Scale bars, 50 μm. (**d**) Percentage of ER-Src cells in S-phase after treatment with EtOH (grey bars) or TAM (orange bars) for 4, 12, 24 or 36 h in the absence of EGF. (**e**) Western blots on protein extracts from ER-Src cells treated with EtOH or TAM for 12 h, blotted with anti-pERK or anti-ERK, and quantification of the ratio of pERK over total ERK levels, normalized to GAPDH for the corresponding lane on western blots. EtOH treatment (grey bar), TAM treatment (orange bar). (**f**) Cyclin D1 mRNA levels on extracts from ER-Src cells treated with EtOH (grey bar) or TAM (orange bar) for 12 h. (**g**) Western blots on protein extracts from ER-Src cells treated with EtOH or TAM and DMSO or MEKi for 12 h, blotted with anti-pERK or anti-ERK, and quantification of the ratio of pERK over total ERK levels, normalized to GAPDH for the corresponding lane on western blots. (**h**) Cyclin D1 mRNA levels on protein extracts from ER-Src cells treated with EtOH or TAM and DMSO or MEKi. (**i**) Number of ER-Src cells in S-phase treated with EtOH or TAM and DMSO or MEKi. DMSO treatment (grey bars), MEKi treatment (blue bars). Quantifications are from (**a**) six or (**c–h**) three biological replicates. Error bars indicate s.d.; NS indicates non-significant; *$P < 0.05$; **$P < 0.001$; ***$P < 0.0001$. Statistical significance was calculated using (**d**) two-way ANOVA or (**e,f**) impaired *t*-test or (**b,g–i**) one-way ANOVA. See also Supplementary Figs 1–3 and 7, and Supplementary Movies 1–4.

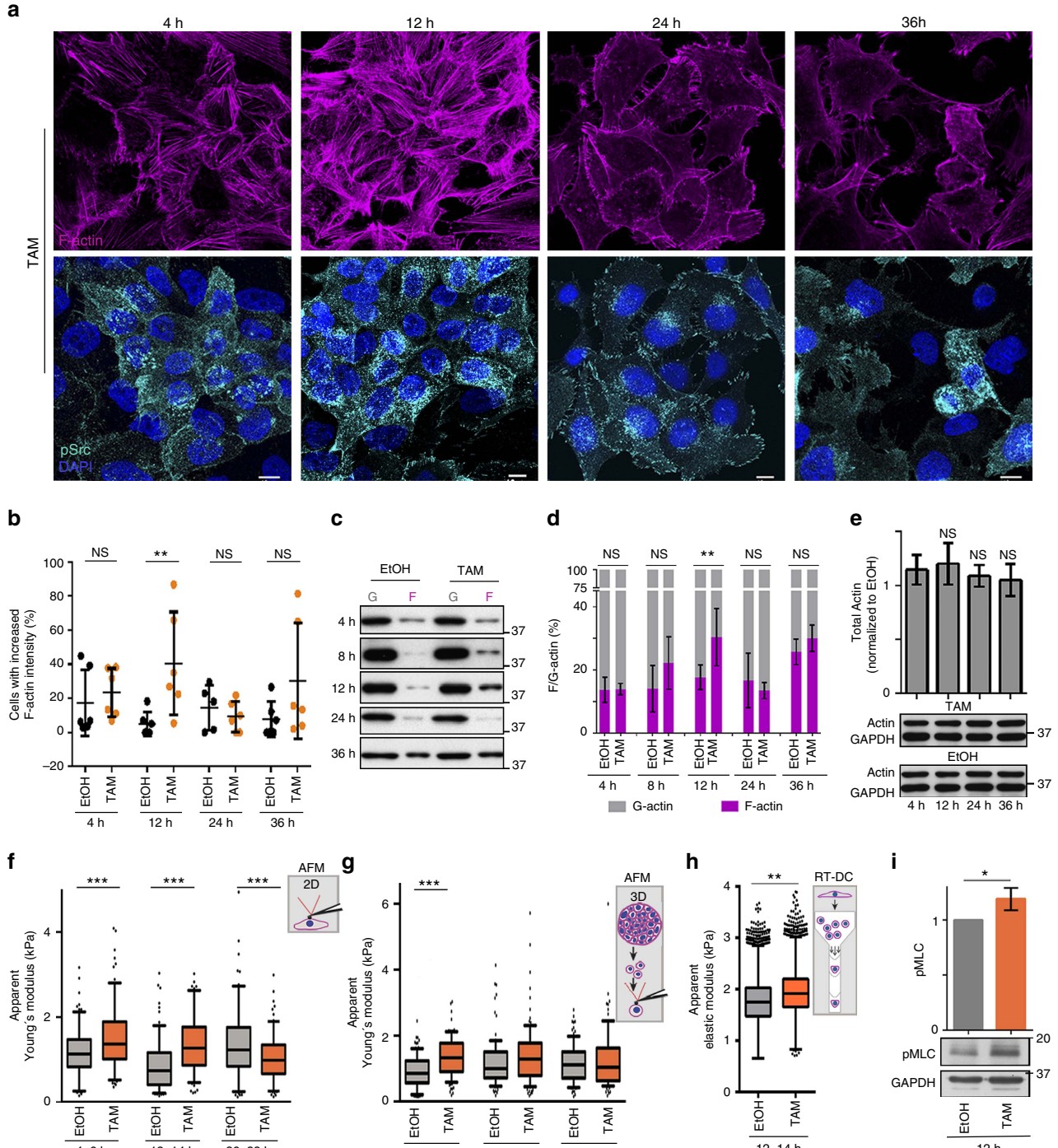

**Figure 2 | Transient F-actin accumulation and cell stiffening by Src activation.** (**a**) Standard confocal sections of ER-Src cells treated with TAM for 4, 12, 24 or 36 h, stained with Phalloidin (magenta) to mark F-actin, anti-p-Src (cyan) and DAPI (blue). Scale bars, 10 μm. (**b**) Percentage of ER-Src cells with high F-actin levels, treated with EtOH (black dots) or TAM (orange dots) for 4, 12, 24 or 36 h. Each dot represents one technical replicate from two biological replicates. Horizontal line indicates median values. (**c**) Western blot on protein extracts from ER-Src cells treated with EtOH or TAM for 4, 8, 12, 24 or 36 h, blotted with anti-actin. (**d**) Quantification from three biological replicates of the G- (grey) and F-actin (magenta) ratio in EtOH- or TAM-treated ER-Src cells for the time points indicated. (**e**) Western blots on protein extracts from ER-Src cells treated with EtOH or TAM for 4, 12, 24 or 36 h, blotted with anti-actin and anti-GAPDH, and quantification from three biological replicates of the ratio of total actin levels between TAM- and EtOH-treated ER-Src cells for the same time points, normalized to GAPDH for the corresponding lanes on western blots. (**f–h**) Apparent Young's moduli of ER-Src cells, treated with EtOH (grey bars) or TAM (orange bars) for the time points indicated, measured by (**f,g**) AFM in **f** 2D or (**g**) 3D, or by (**h**) RT-DC. Data are from three (**f**) or two (**g,h**) biological replicates and are presented as boxplots (25th, 50th, 75th percentiles) and whiskers indicating 10th and 90th percentiles. (**i**) Western blots on protein extracts from ER-Src cells treated with EtOH or TAM for 12 h, blotted with anti-pMLC or anti-GAPDH, and quantification from three biological replicates of pMLC levels, normalized to GAPDH for the corresponding lane on western blots. EtOH treatment (grey bar), TAM treatment (orange bar). Error bars indicate s.d.; NS indicates non-significant; $*P < 0.05$; $**P < 0.001$; $***P < 0.0001$. Statistical significance was calculated using (**b–e**) one-way ANOVA or (**f,g**) simple Mann–Whitney or (**h**) linear mixed models or (**i**) impaired $t$-test. See also Supplementary Fig. 1.

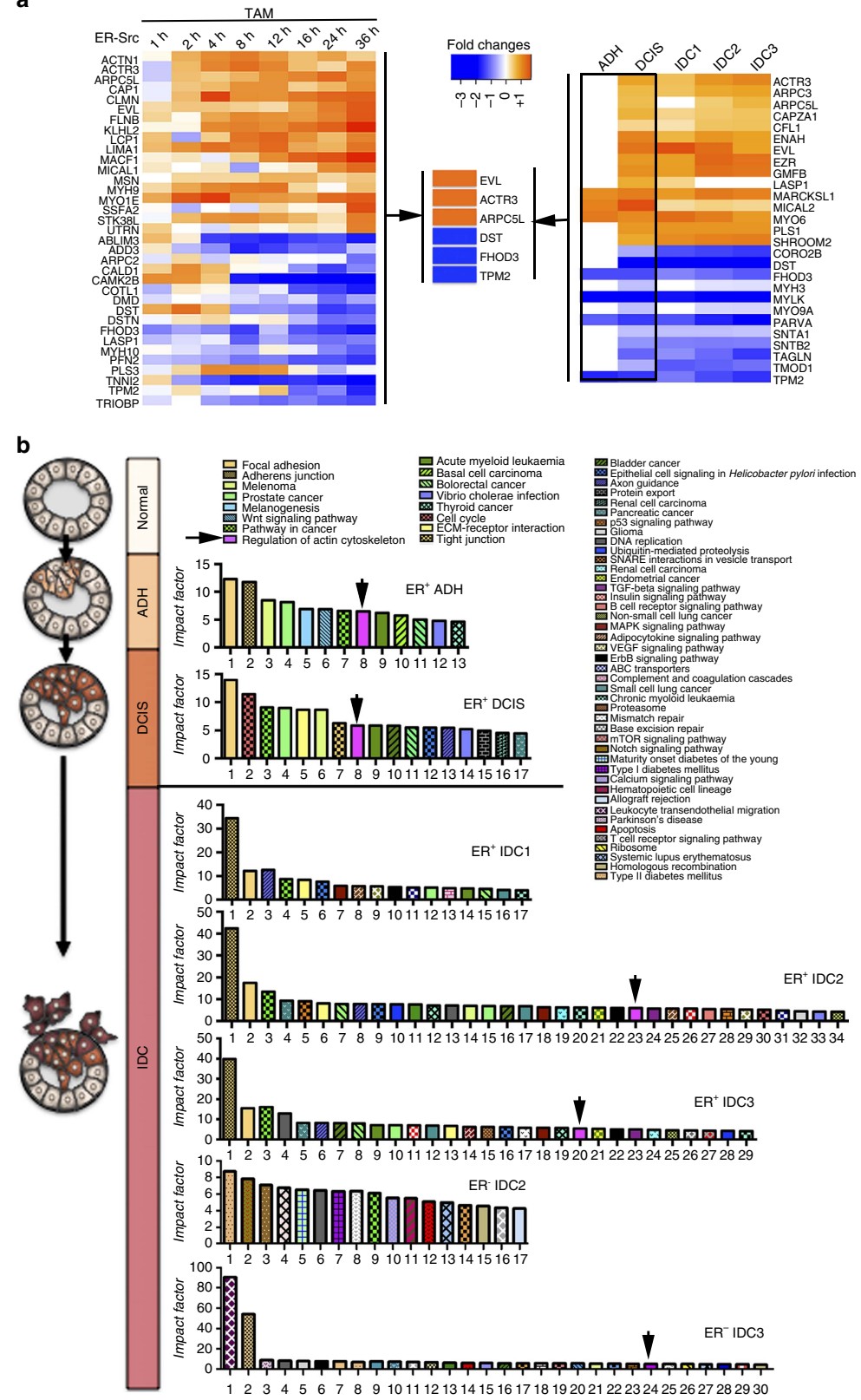

**Figure 3 | Microarray analysis of breast lesions and TAM-treated ER-Src cells.** (**a**) Heatmaps of ABP deregulated in (left panel) ER-Src cells treated with TAM for the different time points indicated[18] or (right panel) ADH and/or DCIS and their expression in the three IDC grades (IDC1, IDC2, IDC3). Fold-changes are in Log2R. (Middle panel) Common ABPs misregulated in the same direction in ADH/DCIS and in TAM-treated ER-Src cells. Orange and blue indicate genes up- and downregulated, respectively. (**b**) (Left) Schematic of the multistep model of breast cancer progression. (Right) Pathway impact analysis for the sets of genes differentially expressed in ER+ ADH, ER+ DCIS, ER+ IDC1, ER+ IDC2, ER+ IDC3, ER− IDC2 and ER− IDC3 when compared to normal breast tissue samples. All pathways significantly affected were plotted on the basis of their impact factor. Black arrows indicate the 'Regulation of actin cytoskeleton' pathway (magenta). See also Supplementary Table 1, Supplementary Data 1 and 2 and Supplementary Fig. 4.

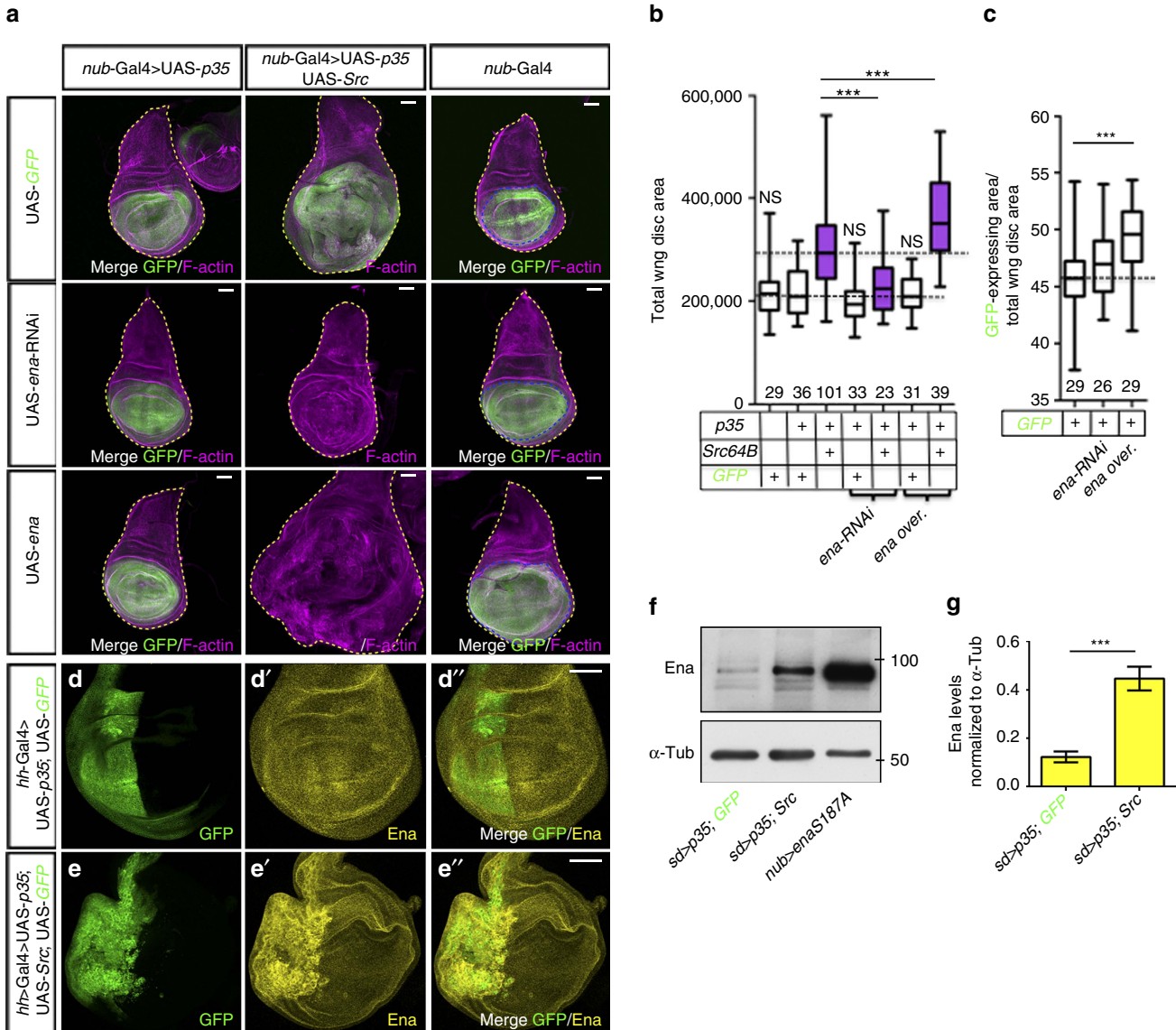

**Figure 4 | *Drosophila* EVL/Ena promotes Src-induced tissue overgrowth. (a)** Standard confocal sections of third instar wing imaginal discs with dorsal side up, stained with Phalloidin (magenta). (Left column) *nub*-Gal4; UAS-*p35* or (middle column) *nub*-Gal4, UAS-*p35*, *Src64B^UY1332^* or (right column) *nub*-Gal4 and carrying (first row) UAS-*mCD8-GFP* (green) or (second row) UAS-*ena-RNAi* or (third row) UAS-*ena*. The yellow dashed lines outline the whole wing disc area. The blue lines outline the Nub > GFP-expressing domains. The scale bars represent 30 μm. **(b,c)** Total wing disc area **(b)** or ratio of the Nub > GFP area over the total wing disc area **(c)** for the genotypes indicated. Numbers of samples indicated on the X axis are from two biological replicates. NS, indicates non-significant for comparison with Nub > GFP,p35. Error bars indicate s.d. ***P < 0.0001. **(d–d″ and e–e″)** Standard confocal sections of third instar wing imaginal discs stained with anti-Ena (yellow), carrying *hh*-Gal4 and **(d–d″)** UAS-*p35* and UAS-*mCD8-GFP* (green) or **(e–e″)** UAS-*p35* and *Src64B^UY1332^* and UAS-*mCD8-GFP* (green). Dorsal side is up and posterior is to the left. Scale bars represent 30 μm. **(f)** Western blots on protein extracts from wing imaginal discs expressing UAS-*mCD8-GFP* and UAS-*p35* (lane 1) or UAS-*p35* and *Src64B^UY1332^* (lane 2) and *sd*-Gal4 or UAS-*enaS187A* and *nub*-Gal4 (lane 3), blotted with anti-Ena and anti-α-Tubulin (α-Tub). **(g)** Quantification from three biological replicates of the ratio of Ena levels normalized to α-Tub for the genotypes indicated. Error bars indicate s.d. ***P < 0.0005. Statistical significance was calculated using the impaired *t*-test. See also Supplementary Fig. 5 and Supplementary Table 2.

ACTR3/Arp3, ARPC5L/Arpc5, DST/Shot, FHOD3/Fhos and TPM2/Tm2 could have the opposite effect.

Quantification of the ratio between the Nub > GFP-expressing area and the total wing disc area showed that overexpressing EVL/Ena (Fig. 4c) or knocking down DST/Shot or TPM2/Tm2 was sufficient to induce the overgrowth of wing discs that did not overexpress Src and p35 (Supplementary Table 2 and Supplementary Fig. 5). Surprisingly, expressing EGFP-Fhos-βB also triggered overgrowth of control Nub > GFP wing discs (Supplementary Table 2 and Supplementary Fig. 5), suggesting

that the effect of Fhos on tissue growth depends on the cellular context. In contrast, overexpressing DST/Shot significantly reduced the growth of these tissues (Supplementary Table 2 and Supplementary Fig. 5). Strikingly, EVL/Ena accumulated in the posterior compartment of wing discs overexpressing Src together with p35 and GFP (Fig. 4e–e″), but not in those expressing p35 and GFP only (Fig. 4d–d″). Accordingly, quantification by western blot indicated that EVL/Ena levels were four times higher in wing discs extracts overexpressing Src and p35 under Scalloped-Gal4 (Sd-Gal4) control, compared

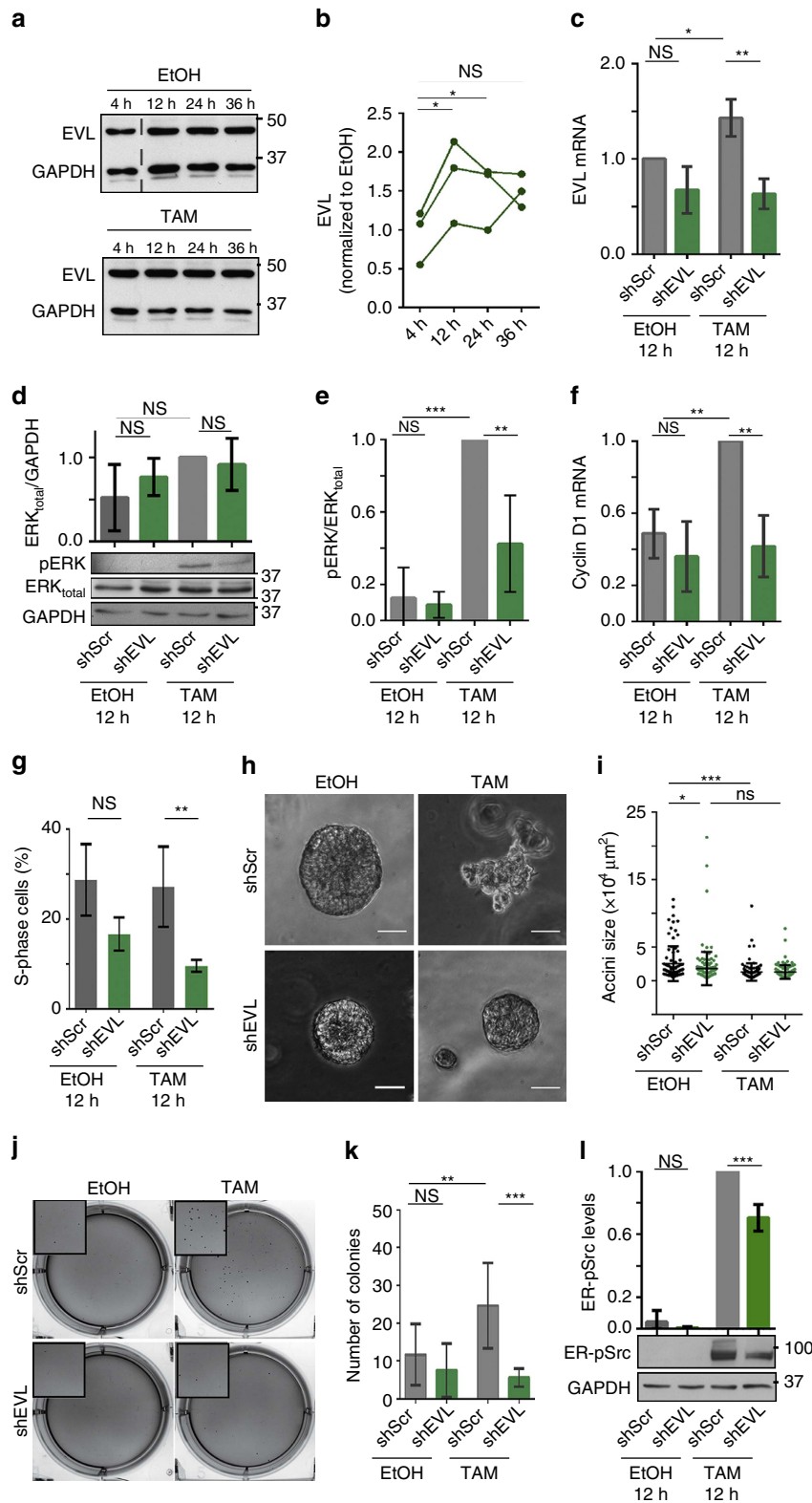

to discs expressing p35 and GFP (Fig. 4f,g). We conclude that *in vivo* EVL/Ena accumulates upon Src activation and is necessary to promote Src-induced tissue overgrowth.

**EVL potentiates ERK-sustained cell proliferation**. We then tested whether EVL was also required for the proliferation of

TAM-treated ER-Src cells. Consistent with the upregulation of EVL on microarrays (Fig. 3), EVL protein transiently accumulated 12 and 24 h after TAM treatment (Fig. 5a,b). Knocking down EVL using two independent short-hairpin RNA (shEVL), significantly reduced EVL mRNA and protein levels, as well as cell growth, in ER-Src cells treated with EtOH or TAM for 36 h in the presence of serum and growth factors, compared to

cells transfected with a scrambled short-hairpin RNA (shScr) (Supplementary Fig. 6). In the absence of serum and growth factors, EVL expression was significantly increased 12 h after TAM treatment in ER-Src cells transfected with shScr (Fig. 5c). This increase in EVL mRNA levels was significant reduced in cells transfected with shEVL. However, EtOH-treated ER-Src cells transfected with shEVL did not show a significant reduction of EVL expression at 12 h (Fig. 5c), suggesting that in the absence of serum and growth factors, ER-Src cells express low EVL levels. In these culture conditions, knocking down EVL in ER-Src cells treated with TAM for 12 h reduced the increase in pERK levels (Fig. 5d,e) without affecting total ERK levels (Fig. 5d) and inhibited the upregulation of Cyclin D1 (Fig. 5f). Knocking down EVL also lowered the number of ER-Src cells in S-phase 12 h after TAM treatment. However, it had no significant effect on cells treated with EtOH for the same period of time (Fig. 5g). Thus, the upregulation of EVL by Src activation potentiates ERK-induced cell proliferation early during cellular transformation.

EVL is also required for TAM-treated ER-Src cells to progress toward a fully transformed phenotype, as knocking down EVL fully suppressed the invasive spike-like phenotype of TAM-treated acini grown in the presence of serum and growth factors for 14 days (Fig. 5h). These acini were significantly smaller than EtOH-treated acini (Fig. 5i), likely due to the fact that TAM-treated cells did not increase their proliferation rate in the presence of serum and growth factors (Supplementary Fig. 2), as well as, extruded from the spherical acini-like structure (Supplementary Movie 4). Consistent with a role of EVL in promoting cell growth in the presence of serum and growth factors (Supplementary Fig. 6), EVL knockdown reduced the size of EtOH-treated ER-Src acini (Fig. 5i). EVL is also required for the tumorigenic potential of Src, as knocking down EVL inhibited the ability of TAM-treated ER-Src cells to produce anchorage-independent colonies in the absence of EGF (Fig. 5j,k). Finally, knocking down EVL in ER-Src cells treated with TAM for 12 h reduced ER-pSrc levels (Fig. 5l), indicating that EVL potentiates Src activity. Altogether these observations demonstrate that EVL is required for Src-dependent ERK activation, which upregulates Cyclin D1 and sustains cell proliferation. In addition, EVL potentiates Src activation and the progression towards a fully transformed phenotype.

**Stress fibres and cell stiffening induced by Src require EVL.** We next tested whether EVL sustains the proliferation of TAM-treated ER-Src cells by assembling the transient Src-dependent actin fibres at 12 h. These F-actin-based structures were largely stained by cytoplasmic β-actin (β-CYA), which predominantly localizes in stress fibres[24], but not by cytoplasmic γ-actin (γ-CYA) (Fig. 6a). Moreover, the Src-dependent actin fibres were associated with larger focal adhesions (FAs), seen by staining with Paxillin (Fig. 6c) and were enriched for pMLC (Fig. 1d), suggesting that they are likely ventral stress fibres. Consistent with a role of EVL in assembling these stress fibres, EVL accumulated at their tips (Fig. 6b). However, knocking down EVL did not suppress the ability of TAM-treated ER-Src cells to form larger FAs at 12 h (Fig. 6c), nor to accumulate F-actin (Fig. 6e,f). On the contrary, EVL knockdown triggered significantly higher amounts of the F-actin pool in EtOH-treated cells (Fig. 6e,f). Quantification of stress fibres anisotropy indicates that 12 h after TAM treatment, ER-Src cells expressing shScr, showed an anisotropic fibre arrangement, compared to those treated with EtOH (Fig. 6g). Strikingly, knocking down EVL in these cells fully suppressed the increase in fibre anisotropy (Fig. 6g). Stress fibre organization by EVL impacts the stiffening of TAM-treated ER-Src cells, as knocking down EVL in these cells prevented their increased cell stiffening at 12 h (Fig. 6h). In addition, knocking down EVL limited the accumulation of pMLC loaded on the Src-dependent stress fibres 12 h after TAM treatment (Fig. 6d). Taken together, we conclude that during the first 12 h of cellular transformation, organization of the Src-dependent stress fibres by EVL promotes cell stiffening, cell proliferation and the progression towards a fully transformed phenotype.

**Cell stiffening potentiates ERK and Src activities.** To test the possibility that the EVL-dependent polarized stress fibres sustain cell proliferation by promoting cell stiffening, we inhibited Myosin II activity in ER-Src cells using Blebbistatin (Blebb). As expected, ER-Src cells co-treated with Blebbistatin and EtOH or TAM for 12 h reduced their stiffness, compared to those treated with DMSO (Fig. 7a). Moreover, Blebbistatin treatment reduced the increase in pERK levels (Fig. 7b,c), as well as the upregulation of Cyclin D1 (Fig. 7d) in ER-Src cells treated with TAM for 12 h in the absence of serum and growth factors. Furthermore, co-treatment of ER-Src cells with Blebbistatin and EtOH or TAM in the presence of serum and growth factors reduced the percentage of S-phase, compared to cells treated with DMSO (Fig. 7e). Finally, Myosin II activity is also required for the stepwise increase of Src activity, as ER-Src cells co-treated with TAM and Blebbistatin for 12 h showed reduced ER-pSrc levels compared to ER-Src cells treated with TAM and DMSO for the same period of time (Fig. 7f). We conclude that Myosin II-dependent cell stiffening is required to potentiate ERK activity and Cyclin D1 expression, and to further enhance Src activity. These observations support a role for the EVL-dependent polarized stress fibres in mediating the proliferative-promoting ability of Src via cell stiffening.

**Figure 5 | EVL sustains Src-induced proliferation via ERK activation.** (**a**) Western blots on protein extracts from ER-Src cells treated with EtOH or TAM for 4, 12, 24 or 36 h, blotted with anti-EVL and anti-GAPDH (see original blot in Supplementary Fig. 6A). (**b**) Ratio of EVL levels between TAM- and EtOH-treated ER-Src cells for the same time points, normalized to GAPDH. (**c**) EVL mRNA levels on extracts from ER-Src cells expressing shScr or shEVL#2 and treated with EtOH or TAM for 12 h. (**d**) Western blots on protein extracts from ER-Src cells expressing shScr or shEVL#2 and treated with EtOH or TAM for 12 h, blotted with anti-pERK or anti-ERK or anti-GAPDH, and quantification of total ERK levels between EtOH- and TAM-treated cells, normalized to GAPDH for the corresponding lanes on western blots. (**e**) Ratio of pERK over total ERK levels, normalized to GAPDH for the experimental conditions indicated. (**f,g**) Cyclin D1 mRNA levels (**f**) or number of cells in S-phase (**g**) for ER-Src cells expressing shScr or shEVL#2 and treated with EtOH or TAM for 12 h. (**h**) 14-day cultures on Matrigel of EtOH- or TAM-treated ER-Src acini, expressing shScr or shEVL#2. Scale bars represent 50 μm. (**i**) Quantification from two biological replicates of acini size for the experimental conditions indicated. Horizontal lines indicate median values. (**j**) Phase-contrast images of ER-Src colonies grown in soft agar without EGF, expressing shScr or shEVL#2 and treated with EtOH or TAM. Inset images correspond to a 135% magnification. (**k**) Number of colonies for the experimental conditions indicated. (**l**) Western blots on protein extracts from ER-Src cells expressing shScr or shEVL#2 and treated with EtOH or TAM for 12 h, blotted with anti-pSrc to reveal ER-pSrc or anti-GAPDH, and quantifications of ER-pSrc levels, normalized to GAPDH for the corresponding lanes on western blots. shScr treatments (grey bars), shEVL treatments (green bars). Quantifications are from three biological replicates, unless indicated. Error bars indicate s.d.; NS indicates non-significant; *$P < 0.05$; **$P < 0.001$; ***$P < 0.0001$. Statistical significance was calculated using (**b**) a Friedman or (**c-l**) one-way ANOVA tests. See also Supplementary Fig. 6.

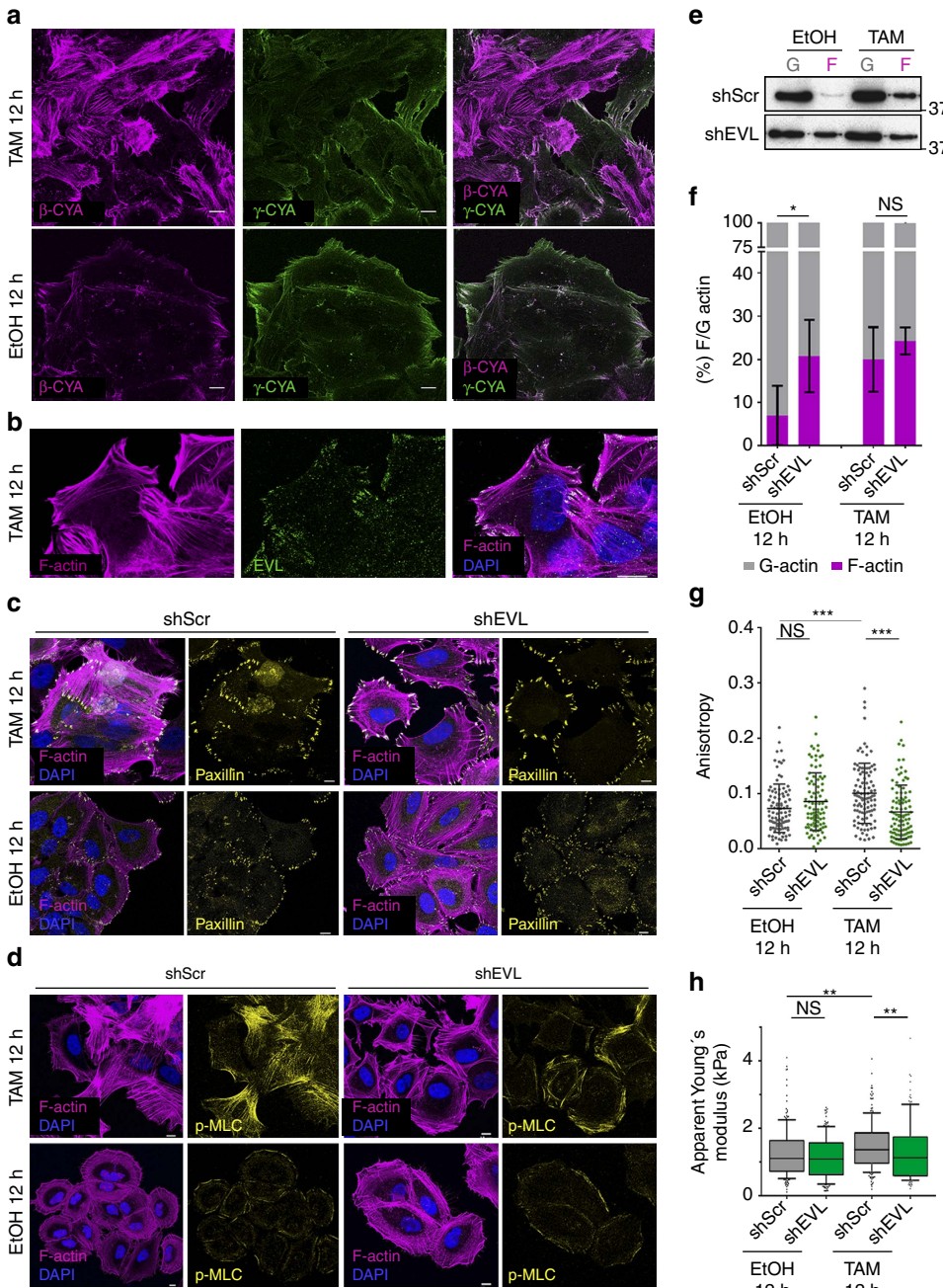

**Figure 6 | The Src-dependent stress fibres and cell stiffening require EVL.** (**a**) Standard confocal sections of ER-Src cells treated with TAM or EtOH for 12 h and stained with β-CYA (magenta) and γ-CYA (green). (**b**) Standard confocal sections of ER-Src cells treated with TAM for 12 h and stained with Phalloidin (magenta), anti-EVL (green) and DAPI (blue). (**c,d**) Standard confocal sections of ER-Src cells treated with TAM or EtOH for 12 h, expressing shScr or shEVL#2, stained with Phalloidin (magenta), DAPI (blue) and (**c**) anti-Paxillin (yellow) or (**d**) anti-pMLC (yellow). All scale bars represent 10 μm. (**e**) Western blot on protein extracts from ER-Src cells treated with EtOH or TAM for 12 h, expressing shScr or shEVL#2, blotted with anti-actin to visualize the G- and F-actin pools. (**f**) Quantification of the G- (grey) and F-actin (magenta) ratio for the experimental conditions indicated. (**g**) Anisotropy of stress fibres in ER-Src cells expressing shScr (grey dots) or shEVL#2 (green dots) and treated with EtOH or TAM for 12 h. Each dot represents the anisotropy of single cells, with the horizontal line indicating median values. (**h**) Apparent Young's moduli of ER-Src cells expressing shScr (grey bars) or shEVL#2 (green bars) and treated with EtOH or TAM for 12 h. Data are presented as boxplots (25th, 50th, 75th percentiles), and whiskers indicate the 10th and 90th percentiles. All quantifications are from three biological replicates. Error bars indicate s.d.; NS indicates non-significant; *$P < 0.05$; **$P < 0.001$; ***$P < 0.0001$. Statistical significance was calculated using (**f,g**) one-way ANOVA or (**h**) the Kruskal–Wallis test.

**EVL accumulates in premalignant DCIS.** To evaluate EVL expression in breast cancer progression, we performed immunohistochemistry in a series of human breast tissues comprising normal breast, DCIS and IDCs. Consistent with its role in F-actin regulation, EVL mainly localized to the cytoplasm in all samples

analysed and could also be observed accumulated at the cell periphery (Fig. 8a and Table 1A). In normal breast, EVL expression was low with only 4.2% (1/24 cases) of the cases being scored as positive and was mainly found in luminal cells. Strikingly, EVL levels were significantly higher in DCIS and IDC

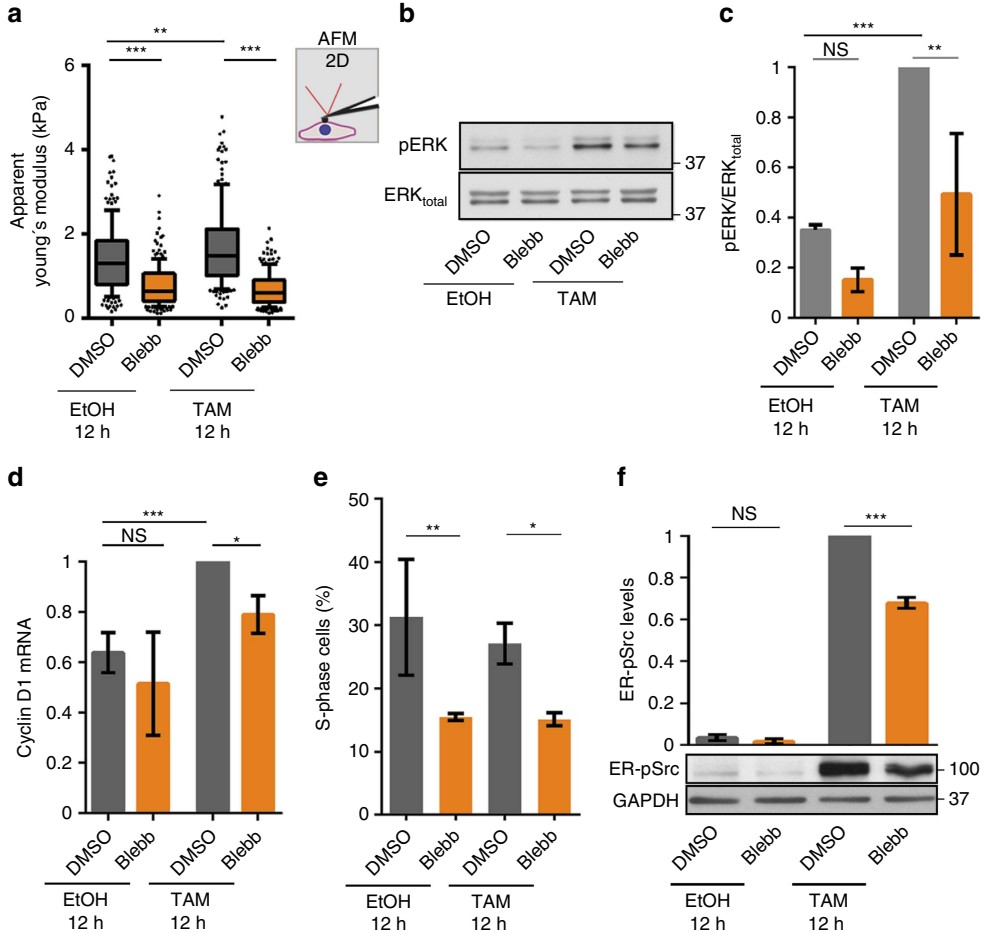

**Figure 7 | Myosin potentiates ERK and ER-Src activities downstream of Src.** (**a**) Apparent Young's moduli of ER-Src cells cultured in 2D, treated with EtOH or TAM and DMSO (grey bars) or Blebbistatin (orange bars) for 12 h. Data are presented as boxplots (25th, 50th, 75th percentiles), and whiskers indicate the 10th and 90th percentiles. (**b**) Western blots on protein extracts from ER-Src cells treated with EtOH or TAM and DMSO or Blebbistatin for 12 h, blotted with anti-pERK or anti-ERK. (**c**) Ratio of pERK over total ERK levels, normalized to GAPDH in ER-Src cells treated with EtOH or TAM and DMSO (grey bars) or Blebbistatin (orange bars) for 12 h. (**d**) Cyclin D1 mRNA levels on protein extracts from ER-Src cells treated with EtOH or TAM and DMSO (grey bars) or Blebbistatin (orange bars) for 12 h. (**e**) Number of ER-Src cells in S-phase treated with EtOH or TAM and DMSO (grey bars) or Blebbistatin (orange bars) for 12 h. (**f**) Western blots on protein extracts from ER-Src cells treated with EtOH or TAM and DMSO or Blebbistatin for 12 h, blotted with anti-pSrc to reveal ER-pSrc or anti-GAPDH, and quantification of ER-pSrc levels, normalized to GAPDH for the corresponding lanes on western blots. All quantifications are from three biological replicates. Error bars indicate s.d.; NS indicates non-significant; *$P < 0.05$; **$P < 0.001$; ***$P < 0.0001$. Statistical significance was calculated using (**a**) simple Mann–Whitney or (**c–f**) one-way ANOVA tests.

lesions, with 56.0% (51/91) and 40.7% (61/150) classified as positive in DCIS and IDC, respectively. Interestingly, IDC presented significantly reduced EVL expression compared to DCIS (Fig. 8a and Table 1A). Indeed, when considering all cases where *in situ* and invasive compartments were found simultaneously on the same slide, we observed a significant reduction in EVL expression from the *in situ* to the invasive counterpart in 54.9% (39/71) of the cases. Only 11.3% (8/71) of the cases showed higher EVL levels in the invasive compartment and 33.8% (24/71) showed no changes in EVL expression (Fig. 8b).

We then investigated in this series of human breast tumours, if the presence of high EVL levels correlates with specific clinicopathological features. Consistent with our microarray analysis of breast tumour samples, in which EVL was found upregulated in ER[+] breast lesions (Supplementary Data 1 and 2), high EVL expression was positively associated with the expression of ER, in both DCIS and IDC, with the luminal A molecular subtype, the absence of epidermal growth factor receptor 2 (HER2) expression, the negative expression of the two basal

markers cytokeratin 5 (CK5) and P-cadherin (P-cad) in DCIS lesions, and with low-grade DCIS and grade I IDC (Table 1B and Supplementary Table 3). Taken together, these observations are consistent with a requirement of high EVL levels for the development of premalignant breast lesions.

## Discussion

In this manuscript, we show that early during cellular transformation, low Src activity promotes stress fibre assembly and upregulates EVL. Stress fibre organization by EVL leads to cell stiffening, ERK activation, Cyclin D1 upregulation, sustained cell proliferation, as well as, enhanced Src activation and the progression towards a fully transformed state. Later during transformation, higher Src activity, in addition to a reduction in EVL levels would disassemble stress fibres to facilitate cell migration (Fig. 9).

In agreement with previously observations[25], Src triggers stress fibre disassembly that would allow for cell migration and

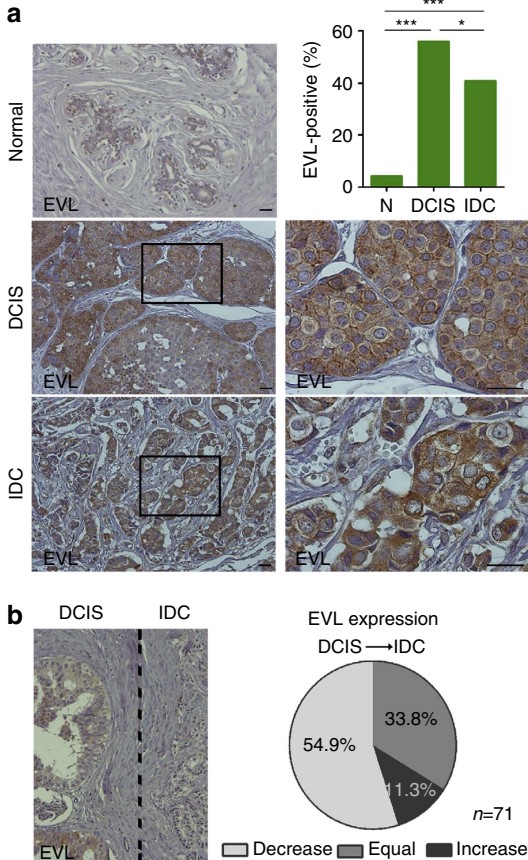

**Figure 8 | EVL predominantly accumulates in DCIS.** (**a**) Representative images of EVL staining by IHC in normal breast tissues, DCIS and IDC, and quantification of EVL protein expression at each stage. *$P = 0.023$; ***$P < 0.0001$. Statistical significance was calculated using Fisher's exact test. Values are from visual scores ± s.e.m. Scale bars represent 100 μm. Images on the right are magnifications of the boxed areas. (**b**) Representative image of EVL staining by IHC from the same breast carcinoma sample containing both an *in situ* and an invasive component and quantification (71 cases). Scale bar represents 100 μm.

invasion. Like in 3D, the acquisition of migrating abilities in monolayer cultures may require the presence of activated Src for 45 h. Alternatively, migrating TAM-treated ER-Src cells could be those that detached from the substratum. Like in invasive cancer cells[8,26], this reduction in F-actin is associated with higher mean deformability. However, our work also reveals that earlier during transformation, Src has opposite effects on stress fibre assembly and cell stiffening, which correlate with the upregulation of Cyclin D1 by ERK activation and the acquisition of proliferative abilities. These cells assemble larger FAs and transient associated-acto-myosin stress fibres, reminiscent to the accumulation of actin-containing microfilament bundles previously reported in chick embryo fibroblasts with conditional Src induction[27]. Accordingly, Src is known to activate the Rho-associated, coiled-coil containing protein kinase (ROCK), which promotes acto-myosin contractility[28]. The apparently opposite effects of Src on stress fibres and cell stiffening correlates with a stepwise increase in Src activity levels and the successive acquisition of premalignant and malignant features in ER-Src cells, as well as in some cancers, such as those of the colon[29–32]. Thus, while low Src activity would promote stress fibre-mediated cell stiffening accompanying tumour growth, higher levels would have the opposite effect, allowing for cell migration.

While Ena/VASP proteins have been proposed to act as anti-capping proteins to permit filament elongation[11], we show here that EVL restricts F-actin accumulation in untransformed ER-Src cells or localizes at the tip of the transient Src-dependent stress fibres to organize them into polarized actin networks and to promote cell stiffening. Thus, EVL may have distinct effects on F-actin depending on the cellular context. Accordingly, several reports describe additional mechanisms by which Ena/VASP proteins regulate F-actin, including preventing branching by the Actin-related protein 2/3 complex (Arp2/3), promoting bundling and recruiting profilin-G-actin complexes[11,33–36]. In untransformed ER-Src cells, EVL may therefore limit Arp2/3 recruitment to F-actin. In contrast, Src-activated cells undergoing transformation may use EVL to cluster stress fibre barbed ends, favouring their bundling along actin filament length. However, we cannot exclude that EVL-dependent nucleation of a sub-population of actin filaments promotes stress fibre organization and cell stiffening. Our observations also argue that stress fibre accumulation *per se,* or an increase in F-actin content, is not sufficient to predict an increase in cell stiffness. As previously proposed[37–39], cellular contractility is also dependent of the spatial organization of stress fibres, with EVL being a key player in this context.

We propose that by supporting acto-myosin contractility, EVL provides a proliferative advantage to cell undergoing transformation. Accordingly, we show that EVL and Myosin II activity potentiates ERK activation, which, in turn, upregulates Cyclin D1 and sustains cell proliferation. ERK overactivation and Cyclin D1 overexpression have been observed in several cancers, including those of the breast, and act as important regulators of tumour cell proliferation[20,40]. Moreover, EVL accumulates predominantly in premalignant breast lesions and its upregulation has been associated with the proliferation of mesothelial cells by glucose degradation product[41]. Furthermore, tumours are frequently detected as rigid masses with cells showing an increase in the elastic modulus resulting from an altered cytoskeleton[42]. Finally, the growth of uterine sarcoma, prostate, lung and breast tumour cells can be prevented using a ROCK inhibitor[43]. Forces applied to stress fibres could directly alter ERK conformation, favouring its activation via phosphorylation. Accordingly, integrin-mediated organization of the actin cytoskeleton regulates ERK nuclear translocation and activity[44,45]. Similarly, mechanical stretch transmitted along the actin cytoskeleton can activate Src[46], justifying the stepwise increase in ER-pSrc levels potentiated by EVL and Myosin II activity and the increase in endogenous pSrc. This positive feedback mechanism on Src would be expected to cause the subsequent decrease in stress fibre-dependent contractility allowing for cell migration. However, we cannot exclude that the effect of inhibiting Myosin II activity on ERK-dependent cell proliferation is independent of the polarized stress fibres by EVL, as Myosin II activity controls several essential cellular processes[47].

Our observations also show that in the presence of serum and growth factors, EVL is required for the growth of untransformed ER-Src cells and suggest that the failure of these cells to proliferate in the absence of growth factors could be due, in part, to the lack of EVL expression. The upregulation of EVL by Src activation would therefore assure that cells acquire self-sufficiency in growth properties. However, how EVL promotes the growth of untransformed ER-Src cells is unclear, as EVL do not appear to control stress fibres organization, nor cell stiffening in these cells.

In addition to EVL, other ABPs might be involved downstream of Src to sustain cell proliferation. DST, FHOD3 and TPM2, may limit tumour growth. Accordingly, a large number of cancers, including those of the breast, downregulate TPM2 (ref. 48).

**Table 1 | EVL predominantly accumulates in luminal A DCIS.**

| A | Count (%) | | | P-values |
|---|---|---|---|---|
| | Normal | DCIS | IDC | |
| *EVL* | | | | |
| Pos | 1 (4.2) | 51 (56.0) | 61 (40.7) | <0.0001 |
| Neg | 23 (95.8) | 40 (44.0) | 89 (59.3) | |
| Total | 24 (100) | 91 (100) | 150 (100) | |

| B | EVL classification—DCIS | | |
|---|---|---|---|
| | Count (%) | | P-values |
| | Neg | Pos | |
| *ER* | | | |
| Neg | 15 (38.5) | 7 (14.0) | 0.008 |
| Pos | 24 (61.5) | 43 (86.0) | |
| Total | 39 (100) | 50 (100) | |
| *HER2* | | | |
| Neg | 29 (74.4) | 48 (96.0) | 0.003 |
| Pos | 10 (25.6) | 2 (4.0) | |
| Total | 39 (100) | 50 (100) | |
| *K5* | | | |
| Neg | 34 (87.2) | 49 (98.0) | 0.043 |
| Pos | 5 (12.8) | 1 (2.0) | |
| Total | 39 (100) | 50 (100) | |
| *P-Cad* | | | |
| Neg | 27 (69.2) | 48 (98.1) | 0.001 |
| Pos | 12 (30.8) | 1 (2.0) | |
| Total | 39 (100) | 50 (100) | |
| *Subtype* | | | |
| Luminal A | 21 (53.8) | 46 (92.0) | 0.002 |
| Luminal B | 5 (12.8) | 1 (2.0) | |
| HER-2 OE | 5 (12.8) | 1 (2.0) | |
| Basal | 6 (15.4) | 1 (2.0) | |
| Ind. | 2 (5.1) | 1 (2.0) | |
| Total | 39 (100) | 50 (100) | |
| *Grade* | | | |
| Low | 10 (25.6) | 20 (40.0) | 0.016 |
| Inter | 11 (28.2) | 21 (42.0) | |
| High | 18 (46.2) | 9 (18.0) | |
| Total | 39 (100) | 50 (100) | |
| *Inflammation* | | | |
| 0 | 1 (5.3) | 7 (30.4) | 0.091 |
| 1 | 13 (68.4) | 13 (56.5) | |
| 2 | 5 (26.3) | 2 (8.7) | |
| 3 | 0 (0.0) | 1 (4.3) | |
| Total | 19 (100) | 23 (100) | |

(A) Number of EVL-positive and -negative cases in normal human breast tissue (normal), DCIS and IDC. Statistical significance was calculated using the $\chi^2$-test. (B) Association of EVL expression with clinicopathological features and breast carcinomas molecular subtypes in DCIS. See also Supplementary Table 3.

Furthermore, restoration of TPM1 and 2 expression in Ras-transformed cells suppresses the transformed phenotype[49]. Surprisingly, while the Arp2/3 complex is known for promoting cancer cell mobility and invasiveness[50], ACTR3 and ARPC5L restrict Src-induced tissue overgrowth *in vivo*. As Arp2/3 controls a multitude of cellular functions[51], it may have opposite functions in tumour growth and invasion/metastasis, respectively. Further studies would be required to confirm the role of these ABPs in tumour growth.

Mouneimne and collaborators have proposed that low EVL levels are predictive of highly invasive tumours and poor prognosis[12]. Accordingly, EVL restricts the migratory ability of MCF10A and untransformed ER-Src cells, as well as SUM159 breast cancer cells[12]. This would justify the need for reducing EVL levels in TAM-treated ER-Src cells that acquire invading abilities[16], as well as in IDCs. However, our observations also reveal that high EVL levels could be a significant predictor of Src-induced breast tumour expansion at earlier stages. Mena11a, another Ena/VASP variant, is also overexpressed in a subset of benign breast lesions associated with HER2 positivity, while downregulated in invasive cells and increases the proliferation rate of MCF-7 cells[52,53]. Thus, several members of the Ena/VASP family could have similar functions in tumour progression. Surprisingly, while Src correlates with ER and HER2 expression and triple negativity[54], EVL is associated with the absence of HER2, the expression of ER and with the luminal A molecular subtype. Thus, EVL may only be required for the growth of a subset of breast tumours that contain high Src activity. Whether in non-Luminal A breast tumours, Src promotes tumour expansion via other ABPs, such as Mena11a is an interesting possibility to be tested in the future. Overall, our work places actin regulation and cell rigidity as central contributors to all stages of in the evolution of breast cancer, which opens new avenues of exploration when designing cancer-targeting therapies.

## Methods

**Normalization and statistical analysis of microarray data.** Raw data (CEL files) from 16 publicly available microarray series were collected from the Gene Expression Omnibus (GEO) database of National Center for Biotechnology Information (NCBI). Both Affimetrix platforms used cover an identical set of 154 genes encoding ABPs. For other genes, only probesets present in both Affimetrix platforms were analysed. R Programming Language for Statistical Computing 2.14.1 (22-12-2011) along with Bioconductor 2.10 packages was used to perform all the calculations[55]. Raw data were subjected to the frozen Robust Multi-Array Average (fRMA) algorithm[56], followed by summarization based on multi-array model, fitted using the Median Polish algorithm. ComBat algorithm[57] was used to decrease the non-biological experimental variation or batch effect for each gene independently. Expression values were transformed in log2, and probesets were mapped using the Entrez gene probeset definition[58]. The Pathway-Express tool from the Onto-tools (http://vortex.cs.wayne.edu) was used to assess biological processes and pathways affected at different stages of ER$^+$ breast tumour samples. Genes encoding for human ABPs were selected from the UNIPROT database on January 2013, using the following key words: 'actin-binding protein'; Organism: '9606'; localized on 'cytoskeleton'; reviewed: 'yes'. The list was verified based on the presence of an actin-binding domain.

**Fly strains and genetics.** Fly stocks used were *nub*-Gal4 (ref. 59), *sd*-Gal4 (ref. 60), Src64B$^{UY1332}$ (ref. 61); UAS-p35 (ref. 62), UAS-arp3-IR$^{KK108951}$, UAS-arpc5-IR$^{KK102012}$, UAS-Tm2-IR$^{KK107970}$, UAS-fhos-IR$^{GD10435}$ (Vienna *Drosophila* Research Center , VDRC), UAS-ena-IR$^{JF01155}$, UAS-enaS187A (ref. 63) UAS-shot-IR$^{GL01286}$, UAS-GFP::arp3 (ref. 64), UAS-ena (ref. 65); UAS-shotL(A)-GFP (ref. 66), UAS-EGFP-fhosΔ$^B$ (ref. 67). All crosses were maintained at 25 °C. Male and female larvae were dissected at the end of the third instar.

**Cell lines, culture conditions and drug treatments.** The MCF10A-ER-Src (ER-Src) and MCF10A-PBabe (PBabe) cell lines were kindly provided by K. Struhl[15]. The presence or absence of the ER-Src fusion construct was confirmed by PCR in the ER-Src and PBabe cells using three independent pairs of primers (5′-Src-ERB GGGAGCAGCAAGAGCAAGCCTAAG and 3′-Src-ERB CGGGGG-TTTTCGGGGGTTGAGC or 5′-Src-ERF GTGGCTGGCTCATTCCCTCACTACA and 3′-Src-ERF GCACCCTCTTCGCCCAGTTGA or 5′-Src-ERE AGAGGGTGC-CAGGCTTTGTG and 3′-Src-ERE GGGCGTCCAGCATCTCCAG). The presence of *Mycoplasma* contamination was tested in both cell lines by PCR using the primers 5′-ACTCCTACGGGAGGCAGCAGT-3′ and 5′-TGCACCATCTGTCAC-TCTGTTAACCTC-3′, which amplifies the spacer regions between the 16S and 23S ribosomal RNA genes. None of the cell lines tested positive for *Mycoplasma* contamination. Cells were grown in a humidified incubator at 37 °C, under a 5% CO$_2$ atmosphere in DMEM/F12 growth medium (Invitrogen, 11039-047), supplemented with 5% horse serum (Invitrogen 16050-122), previously stripped of hormones through dextran-coated charcoal incubation (Sigma C6241), 20 ng per ml of EGF (Peprotech, AF-100-15), 10 μg per ml of insulin (Sigma, I9278), 0.5 μg per ml of hydrocortisone (Sigma, H-0888), 100 ng per ml of cholera toxin (Sigma, C-8052), and penicillin/streptomycin (Invitrogen, 15070-063). To treat cells with 4OH-TAM or EtOH, 50% confluent cells were plated and allowed to adhere for at least 24 h before treatment with 1 μM 4OH-TAM (Sigma, H7904) or with identical

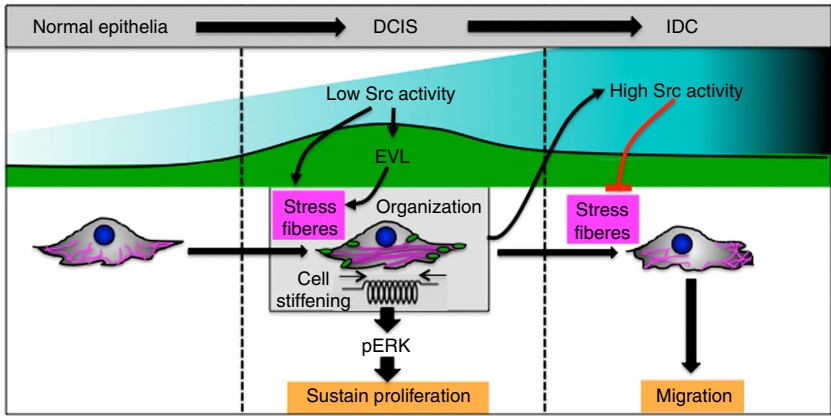

**Figure 9 | Model by which stress fibre and cell stiffening sustain proliferation.** Early during cellular transformation, low Src activity promotes stress fibre assembly and upregulates EVL. Organization of these fibres by EVL leads to cell stiffening. Both EVL and myosin II-dependent cell stiffening potentiates ERK activity, sustains cell proliferation, as well as enhance Src activation. Later during transformation, higher Src activity, in addition to a reduction in EVL levels, would disassemble stress fibres to facilitate cell migration.

volume of EtOH for the time periods indicated in the text. For YAP/TAZ localization assessment, cells were cultured in plain DMEM/F12 during the whole course of the experiments. Plain DMEM/F12 cell culture medium, containing EtOH alone or 4OH-TAM diluted in EtOH, was replaced 4, 8 and 12 h after the beginning of the experiments. To assess ERK activation and Cyclin D1 expression, cells were serum-starved for 16 h in plain DMEM/F12, before treatment with DMSO or 2.5 μM of the MEK inhibitor PD184352 (Sigma, PZ0181) or 10 μM of Blebbistatin (Sigma, B0560), or without any additive for 1 h. Culture medium was then replaced with plain DMEM/F12, containing EtOH or 4OH-TAM, and DMSO or 2.5 μM of PD184352 or 10 μM of Blebbistatin or without any additive, before analysis 12 h later.

**3D matrigel cultures.** Lab-Tek II plates (Lab-Tek II, #155409) were coated with 20 μl Matrigel (BD Biosciences, 356231). A total of 1,000 cells were suspended in 100 μl Matrigel and overlaid in wells. After polymerization at 37 °C, 500 μl of cell culture media was added and replaced every 3 days. Cells were grown for 14 days and treated with cell culture medium containing serum, growth factors and EtOH or 4OH-TAM. Acini were imaged by transmitted light and DIC optics for maximum contrast using a Leica DMI6000 inverted microscope coupled to a Hamamatsu Orca Flash 4.0 sCMOS camera with a Leica 10 × 0.3 numerical aperture PLAN FLUOR dry objective. Image processing and analysis were performed using Fiji (ImageJ package distribution). For size quantification, acini edges were outlined using the Find Edges function, converted to a binary mask before measurement using the area function.

**Time-lapse imaging.** For time-lapse imaging, samples were maintained at 37 °C in a controlled unit with 95% relative humidity, 5% CO$_2$. Transmitted light DIC images were acquired on a Yokogawa CSU-X Spinning Disk confocal scan head, coupled to a Nikon Ti microscope, using a 10 × 0.3 numerical aperture objective, with an Andor iXon + EMCCD camera at 30 or 15 min (min) intervals for a period of 60 h for 3D Matrigel culture or 2D, respectively. Images were acquired using MicroManager 1.4.15 acquisition software. Processing and compression were performed using Fiji. To quantify cell velocity, cell tracking was performed using the Manual Tracking plugin function available in the Fiji software package (quote: http://imagej.net/Citing). Total track length of cell movement was measured and used to quantify the average velocity of each cell per condition.

**Immunofluoresence analysis and quantifications.** For wing imaginal discs, staining was performed by dissecting larvae in phosphate buffer at pH 7 (0.1 M NA$_2$HPO$_4$ and 0.1 M NAH$_2$PO$_4$ at a 72:28 ratio). Discs were then fixed in 4% formaldehyde in PEM (0.1 M PIPES (pH 7.0), 2 mM MgSO$_4$, 1 mM EGTA) for 15–30 min, rinsed in phosphate buffer 0.2% Triton for 15 min and incubated overnight with mouse anti-Ena (1:50, 5G2, Developmental Studies Hybridoma Bank; 1:50) at 4 °C. Discs were then rinsed three times for 10 min in phosphate buffer 0.2% triton, incubated for 1 h with TRITC Donkey anti-mouse (Jackson ImmunoResearch, 715-025-150) in phosphate buffer 0.2% triton supplemented with 10% horse serum and rinsed three more times for 10 min before being mounted in Vectashield (Vector Labs, H-1000). For Phalloidin staining, discs were dissected in phosphate buffer at pH 7 (0.1 M NA$_2$HPO$_4$ and 0.1 M NAH$_2$PO$_4$ at a 72:28 ratio). Discs were then fixed in 4% formaldehyde in PEM (0.1 M PIPES (pH 7.0), 2 mM MgSO$_4$, 1 mM EGTA) for 15–30 min, rinsed in phosphate buffer 0.2% Triton for 15 min and incubated for 1 h at room temperature with Rhodamine-conjugated Phalloidin (Sigma, P-1951) at 0.3 mM in phosphate buffer 0.2% triton X-100 supplemented with 10% horse serum before being rinsed and mounted in Vectashield (Vector Labs, H-1000). Fluorescence images were obtained with an LSM 510 Zeiss or Leica SP5 live confocal microscope using a × 10 or × 20 dry objective. The NIH Image J program was used to quantify wing disc area. Each disc was outlined and measured using the *Area* function, which evaluates size in square pixels. To quantify the ratio of the *nub > GFP* domain over the total wing disc area, the ratio between the area of the GFP domain and the area of the whole disc domain, measured using the *Area* function for each disc, was calculated.

MCF10A cells were plated in poly-L-lysine-coated coverslips (Sigma, P-8920). To stain cells with the anti-β-CYA and anti-γ-CYA, epithelial cell monolayers were fixed with prewarmed 1% paraformaldehyde in DMEM for 30 min, followed by 5 min permeabilization with methanol at − 20 °C. Cells were then rinsed slowly with sequential methanol dilution in phosphate-buffered solution (PBS). For all other fluorescence staining, cells were fixed in 4% paraformaldehyde in PBS at pH 7 for 10 min and permeabilized with TBS-T (TBS—0.1% Triton X-100) at room temperature. For all staining, cells were blocked in TBS-T supplemented with 10% BSA for 1 h at room temperature. Primary antibodies were incubated overnight at 4 °C in blocking solution. Coverslips were then washed three times with TBS and incubated with secondary antibodies and with Rhodamine-conjugated Phalloidin (Sigma, P-1951) at 0.3 mM in blocking solution for 1 h at room temperature. After three washes in TBS, cells were stained with 2 μg ml$^{-1}$ DAPI (Sigma, D9542) for 5 min in TBS, washed again with TBS and mounted on Vectashield. The following primary antibodies were used: anti-EVL (1:50; Sigma, HPA018849), anti-activated Src (1:100; Invitrogen, 44-660G), anti-phospho-Myosin Light Chain 2 (Thr18/Ser19) (1:200; Cell Signaling, 3674), anti-Paxillin (1:200; BD Pharmingen, 610051), anti-β-CYA (1:50; mAb 4C2, IgG1, a gift from C. Chaponnier[24]), anti-γ-CYA (1:100; mAb 2A3, IgG2b, a gift from C. Chaponnier[24]) and anti-YAP (1:200; Santa Cruz Biotechnology, sc101199). The secondary antibodies used to detect anti-β-CYA and anti-γ-CYA were anti-mouse IgG1 FITC-conjugated (1:50; Invitrogen, A21240) and anti-mouse IgG2b alexa 647-conjugated (1:50; Invitrogen, A21141), respectively. The secondary antibodies used to detect all other primary antibodies were IgG TRITC or FITC or Alexa Fluor 647-conjugated (1:200, Jackson Immunoresearch). Fluorescence images were obtained on a Leica SP5 confocal coupled to a Leica DMI6000, using the 63 × 1.4 HCX PL APO CS Oil immersion objective. Quantifications of stress fibres anisotrophy were performed using Fibril Tool plugin for NIH Image J program, where each individual cell was outlined and considered as the region of interest. For quantification of cell fraction with high basal F-actin levels, images were processed and analysed using Fiji. For each cell, the DAPI channel was used to outline the nucleus area. Quantification of F-actin intensity signal was performed for each cell within the nucleus area on projected images of the basal cell surface. The F-actin signal intensity of each cell was then semi-quantitatively scored. Cells were considered negative if their F-actin signal intensity was lower than the average of the five lowest F-actin intensity values. Cells were considered positive if their F-actin signal intensity was higher than three times the average of the five lowest F-actin intensity values. For quantification of YAP/TAZ localization, cytoplasmic localization was defined, as YAP/TAZ was completely absent in the nucleus.

**shRNA adenovirus infection and number of cells.** ER-Src cells were plated in six-well plates to reach 30% confluence by the time of infection. shEVL or shScr virus-containing media were added at a multiplicity of infection of 10$^3$ plaque-forming unit per cell, reaching 100% of efficiency of gene delivery. Media were changed after 24 h. Treatments with cell culture media containing EtOH or 4OH-TAM were performed 72 h after infection.

EVLshRNA#1: 5′-GCCAAATGGAAGATCCTAGTACTCGAGTACTAGGAT-CTTCCATTTGGC-3′;

EVLshRNA#2: 5′-ACGATGACACCAGTAAGAAATCTCGAGATTTCTTAC-TGGTGTCATCG-3′;

ShScr: 5′-GACACGCGACTTGTACCACTTCAAGAGAGTGGTACAAGTC-GCGTGTCTTTTTTACGCGT-3′.

The number of cells for each experimental condition was quantified 36 h after treatment using the Scepter 2.0 Handheld Automated Cell Counter.

**Soft agar colony assay.** Cells ($5 \times 10^3$) in cell culture media containing no EGF were mixed with 0.36% gelling agarose (Sigma, A9045) and plated on top of a solidified layer of 0.7% agarose in cell culture media with no EGF. Cells were fed every 6–7 days with cell culture media with no EGF. The number of colonies was counted 15–21 days later.

**Immunoblotting analysis and quantification.** Protein extracts from MCF10A cells were obtained by scraping cells in TRIS lysis buffer, containing protease (Roche, cOmplete Tablets, 4693159001) and phosphatase (Roche, PhosSTOP Tablets, 4906837001) inhibitors, and lysed for 20 min on ice. Protein extracts from wing imaginal discs were obtained by dissecting discs in phosphate buffer at pH 7 (0.1 M $NA_2HPO_4$ and 0.1 M $NAH_2PO_4$ at a 72:28 ratio). Discs were then lysed in 10 μl 2% SDS. Sample Buffer $2 \times$ was then added to cells or wing imaginal disc lysates before boiling the mixture for 5 min and clearing by centrifugation at 16,168 g at 4 °C for 30 min. Protein was resolved by SDS–PAGE electrophoresis and transferred to PVDF membranes (Amersham Pharmacia, 10600023). Membranes were blocked with 5% milk in TBS 0.1% Tween 20 and incubated with the following: rabbit anti-activated Src (1:1,000; Invitrogen, 44-660G), rabbit anti-EVL (1:250; Sigma, HPA018849), rabbit anti-GAPDH (1:2,000; Santa Cruz, 2D4A7), mouse anti-phospho-p44/42 MAPK (1:2,000; Cell Signaling, 9106), rabbit anti-p44/42 MAPK (1:1,000; Cell Signaling, 9102), rabbit anti-phospho-Myosin Light Chain 2 (Ser19) (1:500; Cell Signaling, 3671), rabbit anti-Actin (1:500; Sigma, A2066), mouse anti-Ena (1:200; Developmental Studies Hybridoma Bank, 5G2) and rabbit anti-α-Tubulin (1:1,000; Sigma, T6199). The specificity of the anti-Ena antibody has been verified on wing disc extracts carrying UAS-enaS187A (ref. 51) and nub-Gal4. Detection was performed by using HRP-conjugated antisera (Jackson Immunoresearch) and Enhanced Chemi-Luminescence (ECL) detection (Thermo Scientific, 32106). Western blots were quantified using the Image Studio Lite program. Uncropped scans of the most relevant western blots can be found in Supplementary Fig. 7.

**Real-time PCR analysis.** One microgram of purified RNA samples was reverse-transcribed using intron–exon-specific primers. Real-time qPCR were performed using PerfeCTa SYBR Green FastMix (Quanta Biosciences) in 384-well plates in the Bio-Rad, #1725125. The relative amount of EVL, Cyclin D1, CTGF or ANKRD1 mRNA was calculated after normalization to the GAPDH transcript. Primers used for EVL were 5′-GAAGAGTCCAACGGCCAGAA-3′ and 5′-ACTGGGAGGCTGCTTTTCTC-3′. Primers used for GAPDH were 5′-CTCTGC-TCCTCCTGTTCGAC-3′ and 5′-ACCAAATCCGTTGACTCCGAC-3′. Primers used for CTGF were 5′-CTCGCGGCTTACCGACTG-3′ and 5′-GGCTCTG-CTTCTCTAGCCTG-3′. Primers used for ANKRD1 were 5′-TAGCGCCCGA-GATAAGTTGC-3′ and 5′-GGTTCAGTCTCACCGCATCA-3′. Primers used for Cyclin D1 were 5′-GATCAAGTGTGACCCGGACTG-3′ and 5′-CCTTGGGGTC-CATGTTCTGC-3′.

**Cell cycle profile.** Sub-confluent cells were cultured overnight in media without EGF. Cells were then incubated in culture media containing EGF and EtOH or 4OH-TAM for 12 h, fixed with 70% EtOH and stained with 10 μg ml$^{-1}$ of propidium iodide (Sigma, P4170). Cell cycle profiles were obtained using FACS Calibur. Flow Logic software was used for quantification of the percentage of S-phase cells.

**G/F actin assay.** The G-actin/F-actin In vivo Assay Kit (Cytoskeleton, Denver, CO, USA, BK037) was used to quantify the F- and G-actin pools. Cells were harvested by scraping in lysis buffer. Cell lysates were then incubated for 10 min at 37 °C and centrifuged at 350 g at room temperature for 5 min to pellet unbroken cells. Supernatants were then centrifuged at 90,000 g for 90 min at room temperature to pellet F-actin and leave G-actin in the supernatant. Supernatants were then removed and kept on the side in $5 \times$ SDS sample buffer, while pellets were incubated in F-actin depolymerization buffer on ice for 1 h to allow actin depolymerization to occur before the addition of $5 \times$ SDS sample buffer. G- and F-actin fractions were then analysed by immunoblotting analysis and quantified using the Image Studio Lite program, as the ratio between F- and G-actin levels for each experimental condition.

**Atomic force microscopy.** For the AFM indentation experiments a Nanowizard I (JPK Instruments, Berlin) equipped with a CellHesion module was used. Arrow-T1 cantilevers (Nanoworld, Neuchatel, Switzerland) were modified with polystyrene beads (radius 2.5 μm, Microparticles GmbH, Berlin, Germany) with the aid of epoxy glue to obtain a well-defined spherical indenter geometry and decrease local strain during indentation. The cantilever was lowered with a speed of 5 μm s$^{-1}$ until a relative set point of 2.5 nN was reached. The resulting force distance curves

were analysed using JPK image processing software (JPK instruments). Force distance data were corrected for the tip sample separation and fitted with a Hertz model for a spherical indenter to extract the apparent Young's Modulus[68]. A Poisson ratio of 0.5 was assumed. A 2-h time window was required for measurement of each experimental condition.

**Real-time deformability cytometry.** A microfluidic chip was made out of PDMS using soft-lithography and sealed with a glass coverslip after plasma surface activation. The microfluidic chip consisted of two reservoirs connected by a 300-μm-long constriction channel with a 30 μm by 30 μm cross-section. A row of filter posts at the inlet prevents the channel from clogging by cell clumps and debris. For measurement, cells were re-suspended in a 0.5% methylcellulose solution in PBS at a concentration of $4–5 \times 10^6$ cells per ml. The cell suspension was drawn into 1 ml syringes and connected to the chip by polymer tubing. Cells were pumped with a syringe pump at a constant flow rate of 0.16 μl s$^{-1}$ (0.04 μl s$^{-1}$ sample + 0.12 μl s$^{-1}$ sheath) for 2 min before collecting data, to stabilize flow. Data (typically, a minimum of 2,000 events) were acquired in real time with a high-speed CMOS camera (MC1362; Mikrotron, Unterschleissheim, Germany), operating at 2,000 f.p.s., and illuminated by a high-power LED (CBT-120, 462 nm; Luminus devices, Woburn, MA, USA) through a $\times 40$ objective of an inverted microscope (Axiovert 200M, Carl Zeiss, Oberkochen, Germany) at the end of the 300-μm-long constriction channel where the cell shape had reached steady state. An image-processing algorithm implemented on C++ LabVIEW was used to determine the cell cross-sectional area and circularity, defined as $c = 2\sqrt{\pi A}/l$, where $A$ is the projected area of the cells and $l$ its perimeter. An analytical model[69] was used to calculate the hydrodynamic flow profile around a moving cell in confinement, couple the resulting surface stress to a linear elastic model of a homogeneous isotropic sphere, and extract the elastic modulus for comparison with the observed cell parameters, assuming a constant viscosity of the surrounding fluid of 15 mPa.

**Breast carcinoma series.** Formalin-fixed and paraffin-embedded breast tumour samples were obtained from a Pathology Laboratory of Araçatuba (Veronese Patologia e Citologia Araçatuba, Brazil), under patient informed consent and with ethical approval by the lab research review boards, implying that the spare biological material, which has not been used for diagnosis, could be used for research. This is in accordance with the national regulative law for the handling of biological specimens from tumour banks, being the samples exclusively available for research purposes in retrospective studies, as well as under the International Helsinki Declaration. All breast tumour cases were previously characterized with regard to ER, progesterone receptor (PR), epidermal growth factor receptor (EGFR), HER2, CK5, P-cad and the antigen Ki-67, in addition to patients' age and clinicopathological features, such as grade, tumour infiltrating lymphocytes (inflammation) and lymph nodes status. Breast tumour components were classified into luminal A and B, HER2-overexpressing, and basal-like carcinomas, according to the immunoprofile[70]. Twenty-four cases of normal breast samples derived from reduction mammoplasties or adjacent to tumour tissue were included in this study. All morphological and immunohistochemical assessments were conducted by a pathologist (AP). The study was conducted under the national regulative law for the handling of biological specimens from tumour banks, the samples being exclusively available for research purposes in retrospective studies.

**Immunohistochemistry and quantification.** IHC was performed using the Ultravision Detection System Anti-polyvalent HRP (Lab Vision Corporation) for EVL. Antigen unmasking was performed using a commercially available solution of citrate buffer, pH 6.0 (Vector Laboratories), at a dilution of 1:100 at 98 °C for 30 min. After the antigen retrieval procedure, slides were washed in PBS and submitted to blockage of the endogenous peroxidase activity by incubation in methanol (Sigma-Aldrich) containing 3% hydrogen peroxide (Panreac). Slides were further incubated with the primary antibody for EVL (1:100; Sigma, HPA018849). The IHC reaction was revealed with diaminobenzidine (DAB) chromogen (DakoCytomation). The positive control was a tonsil sample, which was included in each run, to guarantee the reliability of the assays. Non-neoplastic breast tissues, as well as normal breast surrounding the neoplastic cells, were considered internal controls. EVL expression was mainly found in the cell cytoplasm. The normal breast tissue and each component of breast carcinoma were semi-quantitatively scored with regard to the intensity ($I$) of staining as negative (0, no staining), weak (1, diffuse weak), moderate (2, moderate staining) or strong (3, defined as strong staining), and the extent ($E$) of stained cells was evaluated as a percentage. For each case, a staining score was obtained by multiplying $I \times E$ (range 0-300). Cases with a staining score $\leq 100$ were considered negative and cases with a score $> 100$ were classified as positive. The TMA cores measured 2 mm in diameter and included a representative area of the tumour; up to three cores of each case were represented in the TMA. In the cases where in situ and invasive carcinomas were present in the same core ($n = 71$), the samples were classified using the nomenclature increase/decrease/equal EVL expression according to the change in $I \times E$ score.

**Sample size.** No statistical methods were used to predetermine the sample size. Microarray data of 903 neoplastic breast lesions, including 12 ER$^+$ ADH, 22 ER$^+$

DCIS, 68 ER$^+$ IDC1, 189 ER$^+$ IDC2, 273 ER$^+$ IDC3, 44 ER$^-$ IDC2 and 299 ER$^-$ IDC3, were compared to microarray data of 255 normal breast tissues. For *Drosophila* wing disc growth measurements, 101 samples from four independent *nub*-Gal4, UAS-*GFP* > UAS-*Src64B*$^{UY1332}$, UAS-*p35* crosses (biological replicates) were analysed. For all other crosses, 17–96 samples from two independent crosses (biological replicates) were analysed. Three biological replicates were used for quantification by western blots of pERK, total ERK, pMLC, ER-pSrc, EVL, Ena or actin levels. Six biological replicates were used for quantification of ER-pSrc in ER-Src cells treated with EtOH or TAM for 4, 12, 24 or 36 h. Quantification of EVL protein levels in ER-Src cells knocked down for EVL was carried out from one biological replicate. Protein extracts from wing imaginal discs were obtained by dissecting four discs per genetic background from two independent crosses (biological replicates). To quantify actin levels or F-/G-actin ratio, three biological replicates were used in triplicate (technical replicates). Quantifications of Cyclin D1, EVL, CTGF or ANKRD1 mRNA levels were from three technical replicates, repeated three times (biological replicates). Three biological replicates were used to quantify cell number or number of cells in S-phase of the cell cycle. Quantifications of acini size were from two biological replicates. Quantifications of number of colonies in soft agar were from three biological replicates in triplicate (technical replicates). For quantification of fibre anisotropy, between 90 and 103 cells from three biological replicates were analysed per experimental condition. To quantify the fraction of cells with high basal F-actin levels, 140 to 220 cells from two biological replicates were used. To quantify cell velocity, 100–200 cells from three biological replicates were tracked per condition. To quantify YAP/TAZ localization, eight fields per experimental conditions from two biological replicates were used. A total of 160 (2D) or 100 cells (3D) per experimental conditions from two (2D) or two (3D) biological replicates were used to measure stiffness by AFM. Two biological replicates per condition were used to measure stiffness by RT-DC. The archive of breast tumour samples includes 150 invasive breast carcinomas and 91 *in situ* carcinomas, from which areas of *in situ* and invasive carcinoma can be found in the same block for 71 patients. In addition, 24 normal breast samples were used for comparison.

All experiments were considered for quantification, with the exception of two. For quantification of ER-pSrc levels during the 36 h of cellular transformation, blots with a high degree of variance in GAPDH levels between samples or in ER-pSrc levels between biological replicates were excluded. For quantification of pMLC between ER-Src cells treated with EtOH or TAM for 12 h, three biological replicates, which did not show an increase in ER-pSrc levels and endogenous pSrc, were excluded. Blinding was used to quantify YAP/TAZ localization. No other quantification used blinding. Randomizations were used to quantify fibre anisotropy and the fraction of cells with high basal F-actin levels by selecting each field using the DAPI channel exclusively.

**Statistics.** Statistical analyses of EVL expression in normal, DCIS and IDC samples were conducted using IBM SPSS Statistics (Version 22.0. Armonk, NY: IBM Corp). The associations between categorical variables were tested for statistical significance using the chi-square test to compare multiple conditions or using Fisher's exact tests to compare two conditions. Both tests evaluate the strength of the association between two categorical variables. A two-tailed significance level of 5% was considered statistically significant ($P < 0.05$). Other statistical analyses were performed using Prism 6.0 software (GraphPad Inc.). For statistical comparison of two independent groups (Figs 1e,f,2i and 4b,c,g; and Supplementary Figs 2C and 5) the unpaired *t*-test were used. For statistical comparison between more than two groups (Figs 1a,f,g,2b–d,5b–h,6e,f and 7b–e; Supplementary Figs 2B,3,6B,C,E,F), one-way ANOVA tests were used. For statistical comparison of the percentage of S-phase cells between EtOH- and TAM-treated ER-Src cells over time (Fig. 1d and Supplementary Fig. 2A), two-way ANOVA tests were used. To detect differences in EVL protein levels across multiple test attempts between EtOH- and TAM-treated ER-Src cells during the 36 h of cellular transformation, a Friedman test was used. For statistical comparison of gene expression profiles between normal breast samples and breast tumour samples of different stages, grades and ER status, the LIMMA (Linear Models for Microarray Data) algorithm was used to calculate fold change (Log2R), B-statistics (lods), t-statistics and p-values. Genes selected as differentially expressed were those with a lods value higher than 0: that is, the probability of being differentially expressed was higher that 50% (ref. 71). Statistical significance of apparent Young's moduli between EtOH- and TAM-treated ER-Src cells for each time point (Fig. 2f,g) was calculated using a simple Mann–Whitney test or the Kruskal–Wallis test with multiple comparisons for ER-Src cells expressing shScr or shEVL and treated with EtOH or TAM for 12 h. For RT-DC, statistical analysis was performed using linear mixed models. This approach allows the comparison of different treatment states, considering the reproducibility of the effect. We defined a model that allows an individual mean for each biological replicate (random intercept). In addition, the difference between the compared treatment states (ETOH and TAM) was allowed to be different for each replicate ('random slope'). This model was then compared to a so-called 'Null model', which assumes that there is no difference between the treatment states. A likelihood ratio test and Wilks theorem were used to quantify the similarity of the two models, using a p value. The *P*-value indicates whether the null hypothesis that both models are identical is true.

**Data availability.** Microarray data of genes deregulated in TAM-induced ER-Src cells[18] have been deposited at GEO under accession number GSE17941. Affimetrix platforms used in this study were HG-U133A and HG-U133A PLUS 2.0.; series accession numbers were GSE15852, GSE16873, GSE7390, GSE20194, GSE10810, GSE21422, GSE22544, GSE23593, GSE5460, GSE5764, GSE10780, GSE2109, GSE3744, GSE17907, GSE19615 and GSE23177. See also Supplementary Table 1. All other remaining data are available within the Article and Supplementary Files, or available from the authors upon request.

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

## Acknowledgements

We thank the Bloomington *Drosophila* Stock Centre, the Vienna *Drosophila* Research Center (VDRC) and the Developmental Studies Hybridoma Bank (DSHB) for their services. We specially thank M. Bettencourt-Dias and K. Struhl for reagents, S. Rosa, the IGC ly, Advanced Imaging Unit, Cytometry and Genomics facilities for technical assistance, and C. Chaouiya, R. Oliveira, M. Bettencourt-Dias and C. Lopes for comments on the manuscript. We thank all members of F.J.'s, J.P.'s, J.G.'s and J.B.P.-L.'s labs for helpful discussions. This work was supported by funds from Fundação para a Ciência e Tecnologia (FCT) (PTDC/BIA-BCM/121455/2010) and the Laço Grant in breast cancer 2015 to F.J., and by funds from the Alexander von Humboldt Professorship by the Alexander von Humboldt Foundation to J.G. S.T. was the recipient of fellowships from FCT (SFRH/BD/51884/2012) and Liga Portuguesa contra o Cancro/Pfizer. F.J. is the recipient of IF/01031/2012.

## Author contributions

Conceptualization, S.T. and F.J.; methodology, F.J. and S.T.; investigation, S.T., A.F.V., A.V.T., F.J., N.P.M., C.B.-P., M.A., M.H., O.O. C.B. and A.P.; formal analysis, S.T., A.V.T.; resources, J.C.; writing—original draft, F.J.; writing—review and editing, S.T., J.P., A.F.V. J.G., and A.V.T.; funding acquisition, F.J. and S.T.; supervision, F.J., J.P., J.B.P.-L., J.G. and J.P.
