## [Peer Review File · Nature Communications]

Reviewers' comments:

Reviewer #1 (Remarks to the Author):

This manuscript demonstrates that src induction in a breast cancer cell line initially leads to an increase in cell membrane stiffness and stress fibre accumulation prior to transitioning to a more malignant phenotype. EVL is implicated in regulating this transition and suggests that there may be more than one functional population of actin filaments in the stress fibres which play different roles in the transition and of which a subset (or only 1) may be regulated by EVL. This proposal of a transition from a premalignant stiff state to a softer more malignant state linked to EVL expression is supported by data from both the fly and human breast cancer sections. The core observation makes an important contribution to interpreting the molecular signature of breast cancer biopsy samples and the interpretation of the levels of different markers such as EVL.

Major Comments:

1: The demonstration of the short term impact of src on actin organisation in Figure 1C would benefit from additional quantitation and more representative images. Can the authors quantitate the extent of stress fibre bundles either by using an image analysis program or by calculating the fraction of cells with stress fibre bundles? This is a key observation and needs to be convincingly established.

2: The western blots of F:G actin in Figure 1D do not reflect the associated quantitation. Compare the 36 hr samples with all others. The Western blots suggest that the F:G actin ratio in all time points is dramatically reduced compared to the 36hr samples whereas the quantitation suggests much smaller differences.

3: The variation of the Young's Modulus between the different EtOH time points raises the question of whether the EtOH exposure leads to a transient decrease which src expression nullifies or whether src is increasing stiffness above the initial starting point. Can the different times point be compared with one another and if so, what is the starting value for the Young's Modulus at time 0?

4: Knocking down EVL dissociates stress fibre accumulation from membrane stiffness. Figure 5 demonstrates that EVL knock down in the EtOH control causes both F:G actin (Fig 5C) and stress fibre organisation (Fig 5B) to parallel what is seen in TAM cells but does not impact cell stiffness (scrambled vs EVL knock down) in EtOH cells (Fig 5E). This suggests that it is EVL nucleated actin filaments which regulate the cell stiffness and not the bulk actin filament phenotype, possibly via a specific subpopulation of actin filaments. Based on the data in Figures 5E and 5F it could be argued that an EVL nucleated subpopulation of actin filaments are responsible for the activity of MyoII motors which are regulating cell stiffness. Therefore, care should be taken in equating EVL activity with gross actin organisation.

5: Line 259 states that EVL "could also be observed at the cell membrane (Fig 6A)". At the level of resolution all that can be concluded is that EVL fills the cytoplasm. There is no indication that EVL is really "at the cell membrane" and much higher resolution would be required to draw that conclusion. Similarly it is argued, line 264, that "IDC presented slightly reduced EVL expression compared to DCIS (Fig 6A)". There are no error bars and no statistical test to indicate if the observed difference is significant.

Minor Comments:

1: line 82 should read; ..that prior to cells acquiring malignant...

2: line 169 "role on actin" needs clarification such as 'role in actin filament function' or suchlike.

3: line 311; ...anti-capping proteins...

4: line 318; ...profilin-G-actin....

5: line 329; ...cells...

Reviewer #2 (Remarks to the Author):

The study of Tavares et al., suggests that during the early events of Src-induced cell transformation actin stress-fiber reorganization and cell stiffening mediated by ENA/VASP-like (EVL) promotes cellular proliferation. The authors used the mammary MCF10A model with conditional Src induction, which revealed a transient increase in F-actin and cell stiffness at 12 hrs. Thirty-six hrs after Src induction the fully transformed cells were either more deformable or showed no difference in stiffness compared to control cells. A microarray analysis of Src-activated cells and human breast cancer lesions [pre-malignant atypical ductal hyperplasia (ADH), ductal carcinoma in situ (DCIS) and invasive ductal carcinoma (IDC)] identified a deregulated expression of actin-binding proteins in Src-activated cells as well as in human pre-malignant lesions and DCIS. Interestingly IDC lesions had higher expression levels of genes associated with polymerization-inhibition. Functional studies in drosophilia identified the role of EVL in regulating Src-induced tissue overgrowth. Genetic modulation of Src and EVL expression confirmed the role of EVL in regulating cellular proliferation, the reorganization of stress fibers, cell stiffening and the progression toward the transformed phenotype. Immunohistochemistry was used to measure the expression of EVL in sections from DCIS and IDC lesions.

This article provides some original and significant information on the role of EVL in in vitro cellular transformation and on the functional interaction between Src-activity and EVL. The discovery and experimental approaches used in the study are excellent. However, the weaknesses highlighted below reduce the impact of the study.

It is unclear whether EVL plays a minor or major role in cellular transformation in vivo. The authors did not include in vivo studies assessing the role of EVL in tumor take or growth. It is hard to assess the importance of EVL expression in breast cancer samples and whether the measured difference in EVL expression between DCIS and IDC is really meaningful. In 33.8% of samples there is no difference in EVL expression between DCIS and IDC, and in 11.3% it is higher in IDC than DCIS. Maybe the analysis of p-Src in breast cancer sections would be helpful. Also the quantitative analysis in figure 4A shows a significant increase in EVL at 12 hrs, however at 36 hrs – in the fully-transformed state – the expression of EVL is still high.

While EVL expression promotes cellular proliferation and stiffness, it is unclear whether there is a causal link between cell stiffness and cellular proliferation.

The analysis of EVL expression in sections of breast cancer samples (DCIS, IDC) should be more detailed. The percentage of DCIS and IDC cells expressing EVL should be given. In the samples with both DCIS and IDC, how was the quantitative / statistical analysis performed? What was the minimum number of cells / area of DCIS or IDC required to be included in the analysis?

Reviewer #3 (Remarks to the Author):

Members of the Src-family kinases are conserved non-receptor tyrosine kinases that have been identified as cellular oncogenes promoting tumor cell motility, proliferation, invasion and

metastasis. In this study, Janody and colleagues analyzed the oncogenic effects of Src overexpression and the underlying molecular mechanism of Src-induced cell transformation using an inducible mammary MCF10A epithelial cell culture model and an *in vivo* fly model. The authors found that Src induction leads to an upregulation of EVL, a member of the Ena/VASP protein family of actin regulators. Based on their additional findings the authors propose a model in which malignant morphological features of Src-transformed cells such as actin stress fiber induction, cell stiffening and increased cell growth are caused by EVL upregulation.

Overall, the manuscript is clearly written and most experiments are well documented. While the authors provide novel evidence that Src overexpression act on EVL, the underlying molecular mechanism by which Src acts on EVL at the protein/transcript level remains completely unknown. Even unclear is if all morphological changes of transformed cells are only caused by EVL upregulation or if other targets are involved. Since Src is a non-receptor tyrosine kinase, it also remains unclear whether Src-dependent phosphorylation of EVL or indirectly (e.g. Abl tyrosine kinase) results in actin-dependent changes of cell morphology. The proposed novel role of EVL in promoting actin stress fiber formation and polarization promoting increased cell stiffening is interesting but overall the analysis is not rigorous enough and the data do not support all claims made in the paper. Previous studies already showed that Ena/VASP proteins are upregulated in many human epithelial tumors (including breast, pancreas or colon) promoting their invasiveness, an important phenotype that has been not addressed in this study. Thus, the manuscript in the present version lacks the striking advance to justify publication in Nature Communications. More quantification, a detailed analysis of the Src overexpression effect on EVL and more importantly analysis of how EVL act on the formation/polarization of actin stress fibers would be required.

Specific points:

- 1) The authors stated that cells undergo full phenotypic transformation 36h after TAM treatment with progressive cell detachment. However, the quantification only revealed an increase of 20-30% cells in S-phase of the cell cycle.
- 2) Cells showed no significant difference in the migratory behavior at 12 hours treatment. Are there differences at 4 hours or later? Do the cells show an increased invasive behavior?
- 3) TAM-induced ER-Src cells accumulated actin stress fibers 12 hours after induction. A detailed quantification of this cellular phenotype is required. Are markers for actin stress fibers increased?
- 4) The moderate effect on the G-actin/F-actin ratio only at 12 hours after TAM treatment should be verified by at least three independent experiments. Are there any changes in EVL phosphorylation at 12 hours?
- 5) Cells are significantly stiffer already at 4 hours after TAM treatment before an increase in actin stress fiber formation can be observed. It remains unclear why?
- 6) RT-DC measurements on suspended cells should be also done with cells 4 hours and 36 hours after TAM treatment.
- 7) The authors found a deregulation of 27 ABPs including EVL, Arp2/3 subunits and FHOD3. RNAi-mediated suppression of Arp2/3 and FHOD function in flies however enhanced Src-induced overgrowth, an opposite effect compared to EVL, although the effect of FHOD RNAi seems to be much weaker. Nevertheless, previous studies showed that FHOD formins dramatically promote actin bundle formation in many different systems, in flies as well. Thus, the Src-induced overgrowth in fly imaginal wing discs might be caused by a different mechanism. A gain-of-function experiment using an activated FHOD protein would be important to further support the author's conclusions.
- 8) Endogenous Ena dramatically accumulated in the wing blade upon Src overexpression. The authors should further verify this upregulation in western blots using fly lysates.
- 9) Different from the anti-Ena stainings of wing discs an accumulation of EVL in TAM treated cells is not really obvious, at least much weaker compared to flies. A quantification of western blots from at least three independent experiments should be performed.
- 10) The size of cultured TAM-treated acini compared to controls should be quantified.

Point-by-point response to the referees' concerns.

Reviewer #1 report

This manuscript demonstrates that src induction in a breast cancer cell line initially leads to an increase in cell membrane stiffness and stress fibre accumulation prior to transitioning to a more malignant phenotype. EVL is implicated in regulating this transition and suggests that there may be more than one functional population of actin filaments in the stress fibres which play different roles in the transition and of which a subset (or only 1) may be regulated by EVL. This proposal of a transition from a premalignant stiff state to a softer more malignant state linked to EVL expression is supported by data from both the fly and human breast cancer sections. The core observation makes an important contribution to interpreting the molecular signature of breast cancer biopsy samples and the interpretation of the levels of different markers such as EVL.

Major Comments:

Reviewer 1: point 1: The demonstration of the short term impact of src on actin organisation in Figure 1C would benefit from additional quantitation and more representative images. Can the authors quantitate the extent of stress fibre bundles either by using an image analysis program or by calculating the fraction of cells with stress fibre bundles? This is a key observation and needs to be convincingly established.

Response: To support our observations that Src activation involves the transient accumulation of stress fibers, we now provide in Fig. 2A (previously Fig. 1C) higher resolution images of the transient alteration in F-actin in TAM-treated ER-Src cells. In addition, we have quantified the fraction of cells with increased basal F-actin levels in ER-Src cells treated with EtOH or TAM for 4, 12, 24 and 36 hours. These new observations included in Fig. 2B showed that only treatment with TAM for 12 hours significantly increased the percentage of

cells with higher F-actin levels, compared to cells treated with EtOH for the same period of time (Fig. 2B). We have also compared the levels of pMLC by Western Blot between EtOH- and TAM-treated ER-Src cells at 12 hours. Quantification of 3 biological replicates demonstrates that ER-Src cells treated with TAM for 12 hours contained higher pMLC levels compared to those treated with EtOH for the same period of time (Fig. 2H). Finally, we show in Fig. 6A that the Src-dependent stress fibers were largely stained by cytoplasmic β -actin (β -CYA), which predominantly localizes in stress fibers¹. Taken together, these additional information support our initial observations that Src activation involves the transient accumulation of stress fibers 12 hours after TAM treatment.

Reviewer 1: point 2: The western blots of F:G actin in Figure 1D do not reflect the associated quantitation. Compare the 36 hr samples with all others. The Western blots suggest that the F:G actin ratio in all time points is dramatically reduced compared to the 36hr samples whereas the quantitation suggests much smaller differences.

Response: Quantifications of the ratio between F- and G-actin were from 3 biological replicates, loaded in duplicate. The ratio measures the F-actin signals for each experimental condition, divided by the sum of the G- and F-actin signals (total pool of actin) time 100. We think that the Western Blots shown in the original version of our manuscript (Fig. 2C; previously Fig. 1D) reflect the quantification. We agree with the referee that on Western Blots, ER-Src cells treated with EtOH or TAM for 36 hours appear to contain higher F/G-actin ratio, compared to ER-Src cells treated with EtOH or TAM for 4 or 24 hours or to cells treated with EtOH for 12 hours. This effect could be due to higher cell confluence in both EtOH- and TAM-treated conditions at 36h. These higher F-actin signals at 36 hours appear similar to those of cells treated with TAM for 12 hours. The quantifications reflect these observations. For visualization of Western Blots used to quantify the F-G-actin ratio, please see Fig. 1 of this Rebuttal letter.

Figure 1, this rebuttal letter: Western blots on protein extracts from ER-Src cells treated with EtOH or TAM for 4 (blue), 8 (brown), 12 (red), 24 (yellow) or 36 (green) hours, blotted with anti-actin. Protein extracts are from 3 biological replicates (R1, R2 and R3), loaded in duplicate (a and b). F indicates F-actin. G indicates G-actin.

Reviewer 1: point 3: The variation of the Young's Modulus between the different EtOH time

points raises the question of whether the EtOH exposure leads to a transient decrease which src expression nullifies or whether src is increasing stiffness above the initial starting point. Can the different times point be compared with one another and if so, what is the starting value for the Young's Modulus at time 0?

Response: As noticed by the referee, the apparent Young's Modulus of ER-Src cells treated with EtOH for 12-14 hours is reduced, compared to those of EtOH-treated cells at 4-6 hours or 36-38 hours (Fig. 2E, previously Fig. 1F). We do not know the reason for this effect. However, we do not think that comparing the different time points with one another is very meaningful, as cells are subjected to diverse parameters that are altered over time, such as alterations of the microenvironments (cell confluence; culture medium replacement or others), which could influence cell stiffening. In contrast, 50% confluent cells treated in parallel with EtOH or TAM for the same time period, are subjected to identical alterations in their microenvironment and therefore comparable.

Reviewer 1, point 4: *Knocking down EVL dissociates stress fibre accumulation from membrane stiffness. Figure 5 demonstrates that EVL knock down in the EtOH control causes both F:G actin (Fig 5C) and stress fibre organisation (Fig 5B) to parallel what is seen in TAM cells but does not impact cell stiffness (scrambled vs EVL knock down) in EtOH cells (Fig 5E). This suggests that it is EVL nucleated actin filaments which regulate the cell stiffness and not the bulk actin filament phenotype, possibly via a specific subpopulation of actin filaments. Based on the data in Figures 5E and 5F it could be argued that an EVL nucleated subpopulation of actin filaments are responsible for the activity of MyoII motors which are regulating cell stiffness. Therefore, care should be taken in equating EVL activity with gross actin organisation.*

Response: Although knocking down EVL in ER-Src cells treated with EtOH for 12 hours increased the F/G actin ratio (Fig. 6E, previously Fig. 5C), we did not observe any effect on stress fibers organization in these cells (Fig. 6C, D, previously 5B,F). However, we fully agree with the referee that we cannot exclude a role for EVL-dependent nucleation of a sub-population of actin filaments, which could promote acto-myosin stress fiber assembly and cell stiffening. We now propose this possibility in the discussion. We also agree that we have over-interpreted our observations in the original version of our manuscript by stating that EVL polarizes the Src-dependent stress fibers, as we cannot exclude that EVL is required to assemble this sub-population of stress fiber. However, based on our observations that EVL did not affect the assembly of larger FAs induced by TAM treatment (Fig. 6C), we conclude that EVL organizes the Src-dependent stress fibers. We have replaced "polarization" by "organization" in the text.

Reviewer 1, point 5: *Line 259 states that EVL "could also be observed at the cell membrane (Fig 6A)". At the level of resolution all that can be concluded is that EVL fills the cytoplasm. There is no indication that EVL is really "at the cell membrane" and much higher resolution would be required to draw that conclusion. Similarly it is argued, line 264, that "IDC presented slightly reduced EVL expression compared to DCIS (Fig 6A)". There are no error bars and no statistical test to indicate if the observed difference is significant.*

Response: We agree with the referee that the immunohistochemistry staining of EVL provided in the previous version of our manuscript did not permit to draw any conclusion on EVL sub-cellular localization in breast tumor lesions. We have now included in Fig. 8B (previously 6A) 600 times magnification images showing the cytoplasmic EVL localization, in addition to its sub-cellular accumulation at the cell periphery in DCIS and IDC. The graph in Figure 8B (previously 6A) does not contain error bars, as it is a graphic representation of the percentage of EVL positive cases in normal breast tissues, DCIS and IDC, based in the distribution of the cases. We have now included a table in Figure 8A showing the number of EVL-positive and -negative cases in normal breast tissue, DCIS and

IDC, as well as the result of a chi-square test, which indicates that the number of positive- and negative EVL cases were significantly different between normal breast tissue, DCIS and IDC ($P < 0.0001$). We have also indicated on the graph representing the percentage of EVL-positive cases in Fig. 8B the results of Fisher's exact tests, which indicate that the numbers of EVL-positive IDC ($P = 0.0003$) or DCIS ($P < 0.0001$) were significantly higher when compared to normal breast tissues. Furthermore, this number was significantly reduced in IDC compared to DCIS ($P = 0.023$). These observations are in agreement with a reduction of EVL levels in IDC.

Reviewer 1: Minor Comments:

Reviewer 1: 1: line 82 should read; ..that prior to cells acquiring malignant...

Response: We have replace "...that prior to cells acquire malignant..." by "...that prior to cells acquiring malignant..."

Reviewer 1: 2: line 169 "role on actin" needs clarification such as 'role in actin filament function' or suchlike.

Response: We have replace "role on actin" by "role on F-actin dynamics"

Reviewer 1: 3: line 311; ...anti-capping proteins...

Response: We have replace "actin anti-capping" by "anti-capping proteins"

Reviewer 1: 4: line 318; ...profilin-G-actin....

Response: We have corrected "profilin-G-actin" by "profilin-G-actin"

Reviewer 1: 5: line 329; ...cells...

Response: We have replace "Src activated-cells" by "Src activated cells"

Reviewer #2 report

The study of Tavares et al., suggests that during the early events of Src-induced cell transformation actin stress-fiber reorganization and cell stiffening mediated by ENA/VASP-like (EVL) promotes cellular proliferation. The authors used the mammary MCF10A model with conditional Src induction, which revealed a transient increase in F-actin and cell stiffness at 12 hrs. Thirty-six hrs after Src induction the fully transformed cells were either more deformable or showed no difference in stiffness compared to control cells. A microarray analysis of Src-activated cells and human breast cancer lesions [pre-malignant atypical ductal hyperplasia (ADH), ductal carcinoma in situ (DCIS) and invasive ductal carcinoma (IDC)] identified a deregulated expression of actin-binding proteins in Src-activated cells as well as in human pre-malignant lesions and DCIS. Interestingly IDC lesions had higher expression levels of genes associated with polymerization-inhibition. Functional studies in drosophila identified the role of EVL in regulating Src-induced tissue overgrowth. Genetic modulation of Src and EVL expression confirmed the role of EVL in regulating cellular proliferation, the reorganization of stress fibers, cell stiffening and the progression toward the transformed phenotype. Immunohistochemistry was used to measure the expression of EVL in sections from DCIS and IDC lesions. This article provides some original and significant information on the role of EVL in in vitro cellular transformation and on the functional interaction between Src-activity and EVL. The discovery and experimental approaches used in the study are excellent. However, the weaknesses highlighted below reduce the impact of the study.

Reviewer 2: It is unclear whether EVL plays a minor or major role in cellular transformation in vivo. The authors did not include in vivo studies assessing the role of EVL in tumor take or growth.

Response: We did not assess if knocking down EVL in TAM-treated ER-Src cells inhibited their ability to grow in nude mice. However, we show that Enabled (Ena), the sole *Drosophila* member of the Ena/VASP family accumulates upon Src activation *in vivo* and is necessary to promote Src-induced tissue overgrowth (Fig. 4, previously Fig. 3). Because in this organism, the effects of Src activation are consistent with its effects in mammalian cell culture systems², these observations support a role of EVL downstream of Src activation in tumor growth.

To further support a role of EVL in tumor growth, we search for the mechanism by which EVL could sustain proliferation downstream of Src activation. We tested if EVL could activate signaling through ERK, as this signaling pathway is an important contributor of tumor growth^{3,4} and promotes EGF-independent growth of MCF10A cells⁵, as well as, responds to stress fiber-dependent cell stiffening⁶. We found that pERK levels were significantly higher in ER-Src cells treated with TAM for 12 hours in the absence of serum and growth factors (Fig. 1D). Moreover, these cells upregulated Cyclin D1 (Fig. 1E), a known regulator of G1 to S phase progression⁴. ERK is required to potentiate Cyclin D1 expression in these cells, as co-treatment with the MEK inhibitor PD184352 (MEKi) strongly reduced pERK levels in both EtOH- and TAM-treated cells (Fig. 1F), but prevented the upregulation of Cyclin D1 exclusively in TAM-treated cells (Fig. 1G). Furthermore, in the presence of serum and growth factors, co-treatment of ER-Src cells with MEKi and EtOH or TAM for 12 hours reduced the percentage of S-phase, compared to cells treated with DMSO (Fig. 1H). We conclude that Src activation sustains cell proliferation by potentiating ERK activation and Cyclin D1 upregulation.

In agreement with EVL upregulation being required to sustain proliferation of TAM-treated ER-Src cells, knocking down EVL partly abolished the increase in pERK levels in ER-Src cells treated with TAM for 12 hours in the absence of serum and growth factors without affecting total ERK levels (Fig. 5C) and inhibited the upregulation of Cyclin D1 (Fig. 5D). Knocking down EVL also lowered the number of ER-Src cells in S-phase 12 hours after TAM treatment. However, it had no significant effect on cells treated with EtOH for the same period of time (Fig. 5E). In addition, we show that EVL is required for the stepwise increase of Src activity, as knocking down EVL in ER-Src cells treated with TAM for 12 hours reduced ER-pSrc levels, compared to TAM-treated ER-Src cells transfected with shScr (Fig. 5H). We conclude that EVL is required to promote Src-dependent ERK activation, Cyclin D1 upregulation, cell proliferation, as well as to potentiate Src activation and the progression toward a fully transformed phenotype. Due to the critical role of the Ras-ERK pathway and Cyclin D1 in cancer development and progression^{3,4}, we think that these additional observations support a role of EVL in tumor growth.

Reviewer 2: It is hard to assess the importance of EVL expression in breast cancer samples and whether the measured difference in EVL expression between DCIS and IDC is really meaningful. In 33.8% of samples there is no difference in EVL expression between DCIS and IDC, and in 11.3% it is higher in IDC than DCIS. Maybe the analysis of p-Src in breast cancer sections would be helpful. Also the quantitative analysis in figure 4A shows a significant increase in EVL at 12 hrs, however at 36 hrs – in the fully-transformed state – the expression of EVL is still high.

Response: The scoring of EVL expression in the whole breast series was performed as stated in the Methods section of the manuscript. According to this classification, the cases were scored into positive / negative with a cut-off of Intensity x Extension (IXE) = 100.

Thus, in addition to the graph representing the percentage of EVL-positive cases in Figure 8B (previously Fig. 6A), we have now included a table in Fig. 8A showing the number of EVL-positive and -negative cases in normal breast tissue, DCIS and IDC. Analyzing the whole series, and performing a chi-square test, we found that there was a significant difference in EVL expression between normal breast tissue, DCIS and IDC ($P < 0.0001$). Comparing each group individually, the results of Fisher's exact tests indicate that the numbers of EVL-positive IDC ($P = 0.0003$) or DCIS ($P < 0.0001$) were significantly higher when compared to normal breast tissues. Furthermore, IDC shows significantly less EVL-positive cases, compared to DCIS ($P = 0.0237$), indicating that EVL expression is predominantly enriched in DCIS stage.

To explore if the difference in EVL expression between DCIS and IDC was meaningful within each individual patient, we set out to focus in the 71 cases where both DCIS and IDC components were present on the same slide (Fig. 8C, previously Fig. 6B). For each case, the change in IxE between both components allowed the classification of the cases regarding an increase/decrease/maintenance of EVL expression within the same carcinoma sample. The percentages shown in Figure 8C are indicative of the distribution of cases, so no statistical analysis is presented. This result brings strength to the EVL phenotype observed in the whole series.

As suggested by the referee, we tested if the differences in EVL levels between DCIS and IDC correlate with higher Src activity levels by analyzing the pSrc expression in our series of DCIS and invasive carcinomas. We found that the number of positive cases was very low (only 3 positive cases in a total of 70 in the series of invasive carcinomas, being 2 also positive for EVL; only 1 positive case in a total of 24 DCIS, being also positive for EVL). However, from our experience, the analysis of pSrc usually renders a very small number of positive cases, which may be related with the antibody or the difficulty in the assessment of phosphorylated proteins by IHC. In addition, since there are already cores in the TMA that do not have tumor or have already dropped, the total number of cases is also a limitation to obtain significant associations. We were therefore unable to conclude on whether alterations in EVL levels correlate with alterations in Src activity during breast carcinogenesis.

We agree with the referee that in ER-Src cells, the reduction in EVL levels 36 hours after TAM treatment is not striking (Fig. 5A, previously Fig. 4A). 2D cell tracking did not either show a significant increase in ER-Src cell velocity between 36 and 48 hours after TAM treatment (Supplementary Fig. 2C). Like in 3D (Supplementary Movie 4), the acquisition of migrating abilities in 2D monolayer may also require the presence of activated Src for 45 hours. If so, the reduction in EVL levels could only be initiated 36 hours post-TAM treatment. Alternatively, migrating TAM-treated ER-Src cells could be those that detached from the substratum (Supplementary Movie 2). This pool of cells could contain reduced EVL levels but were discarded for protein extraction. Because our manuscript focuses on a novel role of EVL in sustaining proliferation earlier during cellular transformation, we did not analyze EVL levels in ER-Src cells 48 hours after TAM treatment, nor tested the effect of knocking down EVL on the velocity of these cells. However, our finding, in addition to previous observations from the Brugge's lab showing that EVL restricts the migratory ability of breast cancer cells, and its low expression correlates with bad prognosis⁷, allow us to propose that early during tumor development, high EVL levels could potentiate tumor growth, while a reduction in EVL levels could facilitate cell migration/invasion later during tumor progression.

Reviewer 2: While EVL expression promotes cellular proliferation and stiffness, it is unclear whether there is a causal link between cell stiffness and cellular proliferation.

Response: We agree that the original version of our manuscript did not demonstrate a causal link between cell stiffness and cell proliferation. To establish this link, we inhibited Myosin II activity in ER-Src cells using Blebbistatin. We show now in Fig. 7 that, as expected, ER-Src cells co-treated with Blebbistatin and EtOH or TAM for 12 hours reduced their stiffness,

compared to those treated with DMSO (Fig. 7A). Moreover, Blebbistatin treatment reduced the increase in pERK levels in ER-Src cells treated with TAM for 12 hours in the absence of serum and growth factors (Fig. 7B), in addition to the upregulation of Cyclin D1 (Fig. 7C). Furthermore, co-treatment of ER-Src cells with Blebbistatin and EtOH or TAM in the presence of serum and growth factors reduced the percentage of S-phase, compared to cells treated with DMSO (Fig. 7D). Finally, we show that Myosin II activity is also required for the stepwise increase of Src activity, as ER-Src cells co-treated with TAM and Blebbistatin for 12 hours showed reduced ER-pSrc levels compared to ER-Src cells treated with TAM and DMSO for the same period of time (Fig. 7E). We conclude that Myosin II-dependent cell stiffening is required to potentiate ERK activity and Cyclin D1 expression and to further enhance Src activity. These observations support a role for the EVL-dependent polarized stress fibers in mediating the proliferation promoting abilities of Src via cell stiffening. However, Myosin II activity controls several essential cellular processes⁸. Therefore, we cannot exclude the possibility that the effects of inhibiting Myosin II activity on pERK and ER-pSrc levels are independent of the Src-dependent stress fibers polarized by EVL. We have made this point clear in the discussion and replaced the original title “Actin stress fiber re-organization promotes the growth of pre-invasive breast cancer cells through cell stiffening” by “Actin stress fiber organization promotes cell stiffening and proliferation of pre-invasive breast cancer cells”.

Reviewer 2: The analysis of EVL expression in sections of breast cancer samples (DCIS, IDC) should be more detailed. The percentage of DCIS and IDC cells expressing EVL should be given. In the samples with both DCIS and IDC, how was the quantitative / statistical analysis performed? What was the minimum number of cells / area of DCIS or IDC required to be included in the analysis?

Response: We apologize for not making clear in the original version of our manuscript that EVL expression was scored in DCIS and IDC cases considering the Intensity x Extension. We have now included this information in the Methods section. In sum, normal breast tissue and each component of breast carcinoma were semi-quantitatively scored regarding the intensity (I) of staining as negative (0, no staining), weak (1, diffuse weak), moderate (2, moderate staining) or strong (3, defined as strong staining) and the extent (E) of stained cells was evaluated as a percentage. For each case, a staining score was obtained by multiplying I x E (range 0-300). Cases that presented the staining score = or < 100 were considered negative and the cases with score >100 were classified as positive.

The TMA cores had 2mm of diameter and included a representative area of the tumor and up to 3 cores of each case was represented in the TMA.

In the cases where *in situ* and invasive carcinomas were present in the same core (n=71), the samples were classified using the nomenclature increase/decrease/equal EVL expression according to the change in IxE score. Considering that EVL expression can vary from patient to patient and considering that we only intended to show the distribution of breast carcinoma patients categorized for EVL expression change, no statistical analysis was performed. The analysis of this distribution is represented (in percentage) in Figure 8C.

We observed that 54.9% of the cases showed a decreased in EVL expression from *in situ* to invasive component, supporting the results obtained in the whole series, as well as the *in vitro* data with the TAM-inducible MCF10A-ER-Src cell model.

Reviewer #3 (Remarks to the Author):

Members of the Src-family kinases are conserved non-receptor tyrosine kinases that have been identified as cellular oncogenes promoting tumor cell motility, proliferation, invasion and metastasis. In this study, Janody and colleagues analyzed the oncogenic effects of Src overexpression and the underlying molecular mechanism of Src-induced cell transformation using an inducible mammary MCF10A epithelial cell culture model and an in vivo fly model.

The authors found that Src induction leads to an upregulation of EVL, a member of the Ena/VASP protein family of actin regulators. Based on their additional findings the authors propose a model in which malignant morphological features of Src-transformed cells such as actin stress fiber induction, cell stiffening and increased cell growth are caused by EVL upregulation.

Overall, the manuscript is clearly written and most experiments are well documented. While the authors provide novel evidence that Src overexpression act on EVL, the underlying molecular mechanism by which Src acts on EVL at the protein/transcript level remains completely unknown. Even unclear is if all morphological changes of transformed cells are only caused by EVL upregulation or if other targets are involved. Since Src is a non-receptor tyrosine kinase, it also remains unclear whether Src-dependent phosphorylation of EVL or indirectly (e.g. Abl tyrosine kinase) results in actin-dependent changes of cell morphology. The proposed novel role of EVL in promoting actin stress fiber formation and polarization promoting increased cell stiffening is interesting but overall the analysis is not rigorous enough and the data do not support all claims made in the paper. Previous studies already showed that Ena/VASP proteins are upregulated in many human epithelial tumors (including breast, pancreas or colon) promoting their invasiveness, an important phenotype that has been not addressed in this study. Thus, the manuscript in the present version lacks the striking advance to justify publication in Nature Communications. More quantification, a detailed analysis of the Src overexpression effect on EVL and more importantly analysis of how EVL act on the formation/polarization of actin stress fibers would be required.

Response: We thank the referee for these comments and suggestions, which had greatly improved our manuscript. In this manuscript, we have addressed the role of EVL in sustaining proliferation downstream of Src activation. However, we did not address if a reduction in EVL levels was necessary for these cells to acquire higher migrating ability. The main reason for this is that, although we confirmed previous observations showing that knocking down EVL reduced the velocity of untransformed MCF10A cells (Supplementary Fig. 2B, and ⁷), our interest focuses on actin cytoskeleton's functions in controlling the acquisition of pre-malignant features. We show that EVL is necessary to organize stress fibers downstream of Src activation, leading to transient cell stiffening, ERK-dependent cell proliferation, as well as, to enhance Src activation and the progression towards a fully transformed state. However, our data do not allow us to conclude on whether EVL is sufficient to induce the morphological changes associated to ER-Src cellular transformation 12 hours after TAM treatment. On the contrary, other ABPs regulated by Src activation are likely involved in causing these morphological changes, as EVL did not affect the assembly of larger FAs induced by TAM treatment (Fig. 6C).

Specific points:

Reviewer 3: point 1) *The authors stated that cells undergo full phenotypic transformation 36h after TAM treatment with progressive cell detachment. However, the quantification only revealed an increase of 20-30% cells in S-phase of the cell cycle.*

Response: EtOH-treated ER-Src cell population showed a progressive decrease in the percentage of cells in S-phase of the cell cycle during the 36 hours of treatment. Cells treated with TAM, however, had a significant proliferation advantage over EtOH-treated cells, starting 12 hours after TAM treatment (Fig. 1C, previously Fig. 1B). The proliferative abilities of TAM-treated cells grown in the absence of EGF should be compared to those of control cells treated with EtOH. The quantification in Fig. 1C (obtained from 3 biological replicates) shows that the number of cells in S-phase is in average 2 times higher in TAM-treated cells, compared to those treated with EtOH. At later time points, the difference in proliferative abilities between EtOH- and TAM-treated cells is even higher with TAM-treated cells showing a 4-folds and 6 folds increase in the number of cells in S-phase at 24 or 36 hours, respectively. These observations indicate that TAM-treated cells acquire self-sufficient

in growth properties, a characteristic of transformed cells.

Reviewer 3: point 2) Cells showed no significant difference in the migratory behavior at 12 hours treatment. Are there differences at 4 hours or later? Do the cells show an increased invasive behavior?

Response To investigate the migratory behavior of ER-Src cells, cells were treated with either EtOH or TAM and directly placed under a SP5 Live confocal microscope for time-lapse imaging during the first 12 hours of treatment. Images were captured every hour and cells tracked to determine their velocity. Cell velocity showed no significant difference in the migratory behavior between EtOH- and TAM-treated ER-Src cells during the first 12 hours of treatment (Supplementary Fig. 2A). We apologize for not making clear in our original Figure that velocity was evaluated between 0 and 12 hours of treatments. We have now replaced in the graph showing the quantification of cell velocity, the 12h time point by 0-12 hours and altered the title of Supplementary Fig. 2 accordingly.

As suggested by the referee, we have investigated if ER-Src cells increased their migratory behavior 36 hours after TAM treatment. Cells treated with EtOH or TAM for 36 hours were placed under a SP5 Live confocal microscope for time-lapse imaging and followed during the next 12 hours (36 to 48 hours after treatments). Images were captured every hour and cells tracked to determine their velocity. We now show in Supplementary Fig. 2C that the velocity of TAM-treated ER-Src cells was not significantly higher to the one of EtOH-treated cells after 36 hours of treatment. This might be due to the fact that TAM-treated ER-Src cells that acquire migrating abilities could be those that detach from the substratum (Supplementary Movie 2). Alternatively, although ER-Src cells treated with TAM for 36 hours showed a fully transformed phenotype based on their morphology (Fig. 1B and 2A), cells may only acquire migrating abilities after 48 hours of treatment. In agreement with this possibility, in three-dimensional (3D) cultures of reconstituted basement membrane, cells from TAM-treated acini extruded from the spherical acinar-like structure 45 hours after treatment and invaded the Matrigel (compare Supplementary Movie 3 and 4). We conclude that TAM-induced ER-Src cells acquire self-sufficiency in growth properties prior to migrating abilities.

We did not test the ability of TAM-treated ER-Src cells to invade, as we think that these information would not strengthen our manuscript, which aim to demonstrate a role of stress fibers and cell stiffening downstream of Src in sustaining cell proliferation early during cellular transformation.

Reviewer 3: point 3) TAM-induced ER-Src cells accumulated actin stress fibers 12 hours after induction. A detailed quantification of this cellular phenotype is required. Are markers for actin stress fibers increased?

Response: To support our observations that Src activation involves the transient accumulation of stress fibers, we now provide in Fig. 2A (previously Fig. 1C) higher resolution images of the transient alteration in F-actin in TAM-treated ER-Src cells. In addition, we have quantified the fraction of cells with increased basal F-actin levels in ER-Src cells treated with EtOH or TAM for 4, 12, 24 and 36 hours. These new observations included in Fig. 2B showed that only treatment with TAM for 12 hours significantly increased the percentage of cells with higher F-actin levels, compared to cells treated with EtOH for the same period of time (Fig. 2B). We have also compared the levels of pMLC by Western Blot between EtOH- and TAM-treated ER-Src cells at 12 hours. Quantification of 3 biological replicates demonstrates that ER-Src cells treated with TAM for 12 hours contained higher pMLC levels compared to those treated with EtOH for the same period of time (Fig. 2H). Finally, we show in Fig. 6A that the Src-dependent stress fibers were largely stained by cytoplasmic β -actin (β -CYA), which predominantly localizes in stress fibers¹. Taken together, these additional information support our initial observations that Src activation involves the transient

accumulation of stress fibers 12 hours after TAM treatment.

Reviewer 3: 4) The moderate effect on the G-actin/F-actin ratio only at 12 hours after TAM treatment should be verified by at least three independent experiments. Are there any changes in EVL phosphorylation at 12 hours?

Response: Quantifications of the F/G-actin ratio provided in Fig. 2C (previously Fig. 1D) and Fig. 6E (previously Fig. 5C) were obtained from 3 independent biological replicates (see Fig. 1 of this rebuttal letter). These observations show that ER-Src cells significantly increased their F-actin pool 12 hours after TAM treatment, compared to ER-Src cells treated with EtOH for the same period of time.

Our observations indicate that Src activation upregulated EVL expression in ER-Src cells grown only the absence of serum and growth factors (compare Fig. 5B and Supplementary Fig. 6B,C). In addition, Src activation enhances EVL protein levels in the presence of Serum and growth factors, (Fig. 5A). Thus Src appears to regulate EVL at different levels: transcriptional and protein stability levels, depending on the presence of growth factors. As suggested by the referee, we cannot exclude the possibility that Src also potentiate EVL activity through phosphorylation, either directly or indirectly. We did not test this hypothesis, nor addressed the mechanism by which Src upregulates EVL, as our manuscript focuses on the role of specialized F-actin-based structures in promoting the acquisition of pre-malignant features of cellular transformation. However, we fully agree with the referee that these different levels of EVL regulation by Src are interesting questions to be addressed in the future.

Reviewer 3: point 5) Cells are significantly stiffer already at 4 hours after TAM treatment before an increase in actin stress fiber formation can be observed. It remains unclear why?

Response: The compliance measurements of more than 100 individual cells by AFM for each experimental condition were performed during a 2 hours time window. Therefore, our time point at 4 hours (Fig. 2E, previously Fig. 1F) correspond to measurements of ER-Src cell compliance between 4 and 6 hours after EtOH or TAM treatments. The same apply for the 12 and 36 hours time points. Thus, although ER-Src cells treated with TAM for 4 hours did not show a significant increase in the ratio between F- and G-actin (Fig. 2C, previously Fig. 1D), nor a higher percentage of cells with increased basal F-actin levels (Fig. 2B), the 2 hours time window requirement to perform the AFM measurements could account for the significant increased in cell stiffness already between 4 and 6 hours of TAM treatment. Also, the AFM is highly sensitive to mechanical changes, and we therefore may also pick up F-actin reorganization (causing cell stiffening) before this becomes evident in the analysis of the microscopy images. However, we cannot fully exclude that this initial increase in cell stiffening is independent of stress fiber accumulation, since also cell cortical stiffness contributes to the measured apparent Young's moduli. We apologize for not making clear in the original Figure that these measurements were performed over a 2 hours time period. We have now included in the Method section and in all panels reporting AFM measurements, information concerning the time window required for measurements.

Reviewer 3: point 6) RT-DC measurements on suspended cells should be also done with cells 4 hours and 36 hours after TAM treatment.

Response: As suggested by the reviewer, we have now additionally conducted RTDC measurements after 4 and 36 hours of TAM/EtOH treatments, and also prolonged TAM periods of up to 9 days of treatment. Compared to the AFM experiments, RTDC measurements revealed a somewhat delayed kinetics, with cells becoming always stiffer after TAM treatment for up to 96hrs. Only after prolonged culture times of 9 days the Src-transformed cells were more compliant (Fig. 2, this rebuttal letter), which is in line with the

discussed finding that fully transformed cells are more compliant than non-transformed cells. We interpret these results such that the AFM measurements more closely capture the microscopically observed Src-induced changes in stress fibrils rearrangement, since they probe the cell attached to the surface. In such a situation the mechanical properties will be influenced by cell cortex and the underlying stress fibrils. In contrast, RTDC probes cells in suspension (after their detachment from a 2D culture surface). In these measurements, mostly cell cortical stiffness is probed, and also at much shorter timescales (~1 ms vs. >200ms in AFM; see ⁹). Together, the RTDC measurements may be sensitive to different Src-induced effects on the cytoskeleton, possibly on the actin cortex, which is, however, beyond the scope of this manuscript. Since the interpretation is at this stage rather complex, we are ambivalent if we should still include the RTDC dataset in the manuscript, although it is interesting *per se*. On one hand, we believe that it would be valuable to document our observations with RTDC here (e.g. by providing the full time-course in the supplementary materials together with a short discussion in the legend), since they confirm in principle – importantly with an independent technique- the kinetics of stiffening/softening during oncogenic transformation. On the other hand, showing the dataset would require a more thorough discussion, which may deviate from the main story of this paper. We would like to leave this decision to the reviewers.

Figure 2, this rebuttal letter: TAM-treated ER-Src cells in suspension are stiffer than EtOH-treated cells for the same period of time. Elastic modulus measured by RTDC of suspended ER-Src cells previously cultured in 2D and treated with EtOH or TAM for up to 216 hours.

Reviewer 3:point 7) The authors found a deregulation of 27 ABPs including *EVL*, *Arp2/3* subunits and *FHOD3*. RNAi-mediated suppression of *Arp2/3* and *FHOD* function in flies however enhanced Src-induced overgrowth, an opposite effect compared to *EVL*, although the effect of *FHOD* RNAi seems to be much weaker. Nevertheless, previous studies showed that *FHOD* formins dramatically promote actin bundle formation in many different systems, in flies as well. Thus, the Src-induced overgrowth in fly imaginal wing discs might be caused by a different mechanism. A gain-of-function experiment using an activated *FHOD* protein would be important to further support the author's conclusions.

Response: The Laboratory of Sven Bogdan demonstrated that an activated form of *fhos* (referred to as *knittrig*) promotes actin stress fibers in *Drosophila* macrophages and epithelial tissues ¹⁰. As pointed by the referee, the slight but significantly enhancement of the *Src/p35*-dependent overgrowth when discs were knocked down for *fhos* is surprising (Fig. 3E and J, this rebuttal letter and Supplementary Fig. 5). To determine if overexpressing *fhos* suppresses Src-induced tissue growth, we expressed a UAS-driven N-terminally EGFP-tagged form of *fhos* deleted of the conserved C-terminal basic cluster (*EGFP-fhos-ΔB*), which has been shown to promote F-actin accumulation in third instar wing discs and the assembly of long and thick actin bundles in the salivary gland ¹⁰. Consistent with the effect of knocking down *fhos*, overexpressing *EGFP-fhos-ΔB* suppressed the overgrowth of these tissues (Fig. 3H, J, this rebuttal letter and Supplementary Fig. 5). Because the *fhos-ΔB* construct is fused to *EGFP*, we verified this observation by quantifying the ratio between the *EGFP*-expressing

domain and the total wing area for each disc, which constitutes a more accurate size measurement of the domain affected by the *nub*-Gal4 driver. In agreement with our quantification of total wing area, expressing *EGFP-fhos-ΔB* fully suppressed the overgrowth of *Src/p35*-overexpressing wing discs (Fig. 3K, this rebuttal letter). In addition, *EGFP-fhos-ΔB* severely affected the overall epithelial integrity of *Src/p35*-overexpressing tissues (Fig. 3H, this rebuttal letter). Thus, in the presence of overexpressed Src, Fhos may have multiple effects: it could restrict tissue overgrowth, as well as affect cell-cell adhesion.

Surprisingly, expressing *EGFP-fhos-ΔB* was sufficient to induce the overgrowth of wing discs that did not overexpress *Src* and *p35* (Fig. 3I, L, this rebuttal letter and supplementary Fig. 5). We did not observe any significant effect of expressing *EGFP-fhos-ΔB* on the size of wing discs expressing *p35* and *GFP* (Fig. 3G, J, this rebuttal letter and Supplementary Fig. 5). This could be due to a dosage effect, as *EGFP-fhos-ΔB* is likely expressed at lower levels in *nub>p35*, *GFP* wing discs due to the presence of 2 additional UAS-transgenes. Although these observations suggest that the opposite effects of *fhos* on tissue growth depend on Src activity levels, they are insufficient to draw strong conclusions on the role of Fhos in Src-induced tissue growth. The same apply for Arp2/3 components, Shot and Tm2. All these observations are reported in supplementary data and would require further validations beyond the scope of this manuscript, as our manuscript focuses on the role of EVL/Enabled. We did not either investigated if Src induces the overgrowth of *Drosophila* epithelia through stress fiber accumulation, similar to what we had observed in TAM-induced ER-Src cells. We have included in Supplementary Figure 5 our new data on the effect of expressing *EGFP-fhos-ΔB* on tissue growth and made sure that the conclusions we draw concerning the role of Arp2/3, Fhos, Shot and Tm2 in Src-induced tissue overgrowth were not over-interpreted.

Reviewer 3 point 8) Endogenous *Ena* dramatically accumulated in the wing blade upon *Src* overexpression. The authors should further verify this upregulation in western blots using fly lysates.

Response: We have now verified by Western Blot that *Ena* accumulates in wing discs overexpressing *Src*. We have included these new observations in Fig 4H. We show that *Ena* levels are 4 times higher in wing discs extracts overexpressing *Src* and *p35* under *scalloped-Gal4* (*sd-Gal4*) control, compared to wing disc extracts expressing *p35* and *GFP*. To control for the specificity of the anti-*Ena* antibody, we have also loaded wing disc extracts overexpressing UAS-*enaS187A*¹¹ under *nub-Gal4* control. Quantifications were obtained from 3 independent experiments.

Reviewer 3 point 9) Different from the anti-*Ena* stainings of wing discs an accumulation of *EVL* in TAM treated cells is not really obvious, at least much weaker compared to flies. A quantification of western blots from at least three independent experiments should be

performed.

Response: Our comparison of EVL/Ena protein levels in MCF10A cells and in *Drosophila* wing discs upon Src activation indicate that while we observed a 2-folds EVL increased in average in ER-Src cells treated with TAM for 12 hours (Fig. 5A; quantified on Western Blots from 3 independent biological replicates), *Drosophila* Ena increased 4-folds upon *Src/p35*-overexpression (Fig. 4H). We did not investigate the reason for this difference. This could be simply due to differences in Src expression or activity levels in each model.

Reviewer 3 point 10) The size of cultured TAM-treated acini compared to controls should be quantified.

Response: We have now quantified the size of EtOH and TAM-treated acini. We show in Fig. 5F that TAM-treated acini grown in the presence of serum and growth factors for 14 days, were significantly smaller than EtOH-treated acini, This was expected for two reasons. First, cells from TAM-treated acini extruded from the spherical acinar-like structure and invaded the Matrigel (compare Supplementary Movie 3 and 4), which would lead to a reduction of the number of cells per acini. Second in the presence of serum and growth factors, including EGF, EtOH- and TAM-treated ER-Src cells proliferated at similar rate during the 36 hours of treatment (Fig. 4, this rebuttal letter and Supplementary Fig. 2A). The proliferation advantage of TAM-treated ER-Src cells was only observed in the absence of EGF (Fig. 4, this rebuttal letter and Fig. 1C, previously Fig. 1B). The same behavior has been observed upon activation of other oncogenes in cultured cells. Thus, like TAM-treated ER-Src cells, YAP overexpression does not affect the growth rate of MCF10A in the presence of EGF, but sustain proliferation of these cells in the absence of EGF¹². This indicates that in cultured cells, while EtOH-treated ER-Src cells (Fig. 1C) or MCF10A cells carrying a control vector¹² absolutely need EGF for cell proliferation, TAM-treated ER-Src cells (Fig. 1C) or MCF10A-ER-Src cells overexpressing YAP¹² gain the ability to sustain proliferation without EGF. By testing the effect of knocking down EVL in TAM-treated acini (Fig. 5F, previously Fig. 4E), we aimed at investigating if EVL affects the progression toward a fully transformed phenotype. We show that knocking down EVL fully suppressed the invasive spike-like phenotype of TAM-treated acini (Fig. 5F), suggesting that by sustaining proliferation of TAM-treated cells early during cellular transformation, EVL triggers the progression toward a fully transformed phenotype. We have now included in Supplementary Fig. 2 the proliferation rates of EtOH- and TAM-treated ER-Src cells in the presence of EGF and clarified the text.

Figure 4, this rebuttal letter: EGF supports the proliferation of EtOH-treated ER-Src cells. Percentage of ER-Src cells in S-phase grown in the absence (A) or presence (B) of EGF, after treatment with EtOH (grey bars) or TAM (orange bars) for 4, 12, 24 or 36 hours.

We have also quantified the size EtOH- and TAM-treated acini, knocked down for EVL. We show in Fig. 5F that EVL knocked down only reduced the size of EtOH-treated ER-Src acini grown in complete medium for 14 days. In 2D culture, EVL is also required for the growth of EtOH-treated ER-Src cells grown in complete medium (Supplementary Fig. 6E,F). Because in the absence of serum and growth factors, EVL appeared to be expressed at low

levels (Fig. 5B), the failure of untransformed ER-Src cells to proliferate in these conditions could be, in part, due the lack of EVL expression. The upregulation of EVL by Src activation would therefore assure that cells acquire self-sufficiency in growth properties. However, how EVL promote the growth of un-transformed ER-Src cells is unclear, as EVL do not appear to control stress fibers organization, nor cell stiffening in these cells.

References

- ¹ Dugina, V. et al., Beta and gamma-cytoplasmic actins display distinct distribution and functional diversity. *J Cell Sci* **122** (Pt 16), 2980 (2009).
- ² Vidal, M., Larson, D. E., and Cagan, R. L., Csk-deficient boundary cells are eliminated from normal Drosophila epithelia by exclusion, migration, and apoptosis. *Dev Cell* **10** (1), 33 (2006); Vidal, M., Warner, S., Read, R., and Cagan, R. L., Differing Src signaling levels have distinct outcomes in Drosophila. *Cancer Res* **67** (21), 10278 (2007); Fernandez, B. G., Jezowska, B., and Janody, F., Drosophila actin-Capping Protein limits JNK activation by the Src proto-oncogene. *Oncogene* (3013).
- ³ Maik-Rachline, G. and Seger, R., The ERK cascade inhibitors: Towards overcoming resistance. *Drug Resist Updat* **25**, 1 (2016).
- ⁴ Neuzillet, C. et al., MEK in cancer and cancer therapy. *Pharmacol Ther* **141** (2), 160 (2014).
- ⁵ Debnath, J., Walker, S. J., and Brugge, J. S., Akt activation disrupts mammary acinar architecture and enhances proliferation in an mTOR-dependent manner. *J Cell Biol* **163** (2), 315 (2003).
- ⁶ Aplin, A. E., Stewart, S. A., Assoian, R. K., and Juliano, R. L., Integrin-mediated adhesion regulates ERK nuclear translocation and phosphorylation of Elk-1. *J Cell Biol* **153** (2), 273 (2001); Aplin, A. E. and Juliano, R. L., Regulation of nucleocytoplasmic trafficking by cell adhesion receptors and the cytoskeleton. *J Cell Biol* **155** (2), 187 (2001).
- ⁷ Mouneimne, G. et al., Differential remodeling of actin cytoskeleton architecture by profilin isoforms leads to distinct effects on cell migration and invasion. *Cancer Cell* **22** (5), 615 (2012).
- ⁸ Newell-Litwa, K. A., Horwitz, R., and Lamers, M. L., Non-muscle myosin II in disease: mechanisms and therapeutic opportunities. *Dis Model Mech* **8** (12), 1495 (2015).
- ⁹ Mietke, A. et al., Extracting Cell Stiffness from Real-Time Deformability Cytometry: Theory and Experiment. *Biophys J* **109** (10), 2023 (2015).
- ¹⁰ Lammel, U. et al., The Drosophila FHOD1-like formin Knittrig acts through Rok to promote stress fiber formation and directed macrophage migration during the cellular immune response. *Development* **141** (6), 1366 (2014).
- ¹¹ Lucas, E. P. et al., The Hippo pathway polarizes the actin cytoskeleton during collective migration of Drosophila border cells. *J Cell Biol* **201** (6), 875 (2013).
- ¹² Overholtzer, M. et al., Transforming properties of YAP, a candidate oncogene on the chromosome 11q22 amplicon. *Proc Natl Acad Sci U S A* **103** (33), 12405 (2006).

REVIEWERS' COMMENTS:

Reviewer #1 (Remarks to the Author):

I am satisfied that the authors have provided convincing answers to all the points raised in my review.

Peter Gunning

Reviewer #3 (Remarks to the Author):

The manuscript is improved. The authors have addressed my specific points/comments adequately. It is a shame that the authors did not provide further insights into the molecular mechanism by which Src acts on EVL and/or how EVL organizes Src-dependent stress fibers. However, the new data on Myosin II activity further provide strong evidence that Myosin II-dependent cell stiffening is required to potentiate ERK activity and Cyclin D1 expression, thus promoting cell proliferation. Therefore, I am happy to recommend publication of this manuscript.

Minor point: It remains unclear why the authors used a phosphomutant (EnaS187A) instead of a wild type Ena protein to verify anti-Ena antibody specificity.

Point-by-point response to the referees' concerns.

Reviewer 1: *I am satisfied that the authors have provided convincing answers to all the points raised in my review.*

Response: We thank the referee for his appreciation of our work and for his useful recommendations, which were essential to improve our manuscript.

Reviewer #3 (Remarks to the Authors)

Reviewer 3: *The manuscript is improved. The authors have addressed my specific points/comments adequately. It is a shame that the authors did not provide further insights into the molecular mechanism by which Src acts on EVL and/or how EVL organizes Src-dependent stress fibers. However, the new data on Myosin II activity further provide strong evidence that Myosin II-dependent cell stiffening is required to potentiate ERK activity and Cyclin D1 expression, thus promoting cell proliferation. Therefore, I am happy to recommend publication of this manuscript.*

Response: We thank the referee for his appreciation of our work and for his useful recommendations, which were essential to improve our manuscript. We agree with the referee that our manuscript leave opened many interesting questions, including how Src controls EVL expression and/or activity and how, in turn, EVL organizes the Src-dependent stress fibers. We are aiming to tackle these questions in the near future.

Reviewer 3: *Minor point: I remains unclear why the authors used an phosphomutant (EnaS187A) instead of a wild type Ena protein to verify anti-Ena antibody specificity.*

Response: The reviewer is right. We had no specific reasoning for using a phospho-mutant form of Ena (EnaS187A), instead of wild type Ena to verify the specificity of the anti-Ena antibody. Our fly stock collection originally included three UAS-*ena* expressing lines, two of them driving expression of wild type *ena*^{1,2}, and one driving a form of *ena* insensitive to phosphorylation by Warts (*enaS187A*)². The wild type UAS-*ena* stock provided by Barry Thompson, inserted at *attP 28E7* on the second chromosome showed signs of contamination, seen by variability in eye colors, while the other wild type UAS-*ena* line was very weak. Because of time constrain, we instead used the UAS-*enaS187A* stock we had available. Nevertheless, we think that this form of Ena is also appropriate to verify that the strongest band observed in Src-overexpressing wing disc migrate at similar molecular weight on Western Blot than EnaS187A.

Reference

1. Ahern-Djamali, S.M. et al. Mutations in *Drosophila* enabled and rescue by human vasodilator-stimulated phosphoprotein (VASP) indicate important functional roles for Ena/VASP homology domain 1 (EVH1) and EVH2 domains. *Mol Biol Cell* **9**, 2157-2171 (1998).
2. Lucas, E.P. et al. The Hippo pathway polarizes the actin cytoskeleton during collective migration of *Drosophila* border cells. *J Cell Biol* **201**, 875-885 (2013).